# Fair Cooperation in Mixed-Motive Games via Conflict-Aware Gradient Adjustment

**Woojun Kim**
Robotics Institute
Carnegie Mellon University
Pittsburgh, PA 15213
woojunk@andrew.cmu.edu

**Katia Sycara**
Robotics Institute
Carnegie Mellon University
Pittsburgh, PA 15213
sycara@andrew.cmu.edu

## Abstract

Multi-agent reinforcement learning in mixed-motive settings presents a fundamental challenge: agents must balance individual interests with collective goals, which are neither fully aligned nor strictly opposed. To address this, reward restructuring methods such as gifting and intrinsic motivation have been proposed. However, these approaches primarily focus on promoting cooperation by managing the trade-off between individual and collective returns, without explicitly addressing fairness with respect to agents' task-specific rewards. In this paper, we propose an adaptive conflict-aware gradient adjustment method that promotes cooperation while ensuring fairness in individual rewards. The proposed method dynamically balances policy gradients derived from individual and collective objectives in situations where the two objectives are in conflict. By explicitly resolving such conflicts, our method improves collective performance while preserving fairness across agents. We provide theoretical results that guarantee monotonic non-decreasing improvement in both the collective and individual objectives and ensure fairness. Empirical results in sequential social dilemma environments demonstrate that our approach outperforms baselines in terms of social welfare, while maintaining fairness.

## 1 Introduction

Multi-agent reinforcement learning (MARL) aims to train multiple agents to maximize cumulative rewards in a given task. Depending on the reward structure, MARL is typically categorized into three settings: cooperative, adversarial, and mixed-motive. In the mixed-motive setting, agents' rewards are neither fully aligned (as in cooperative settings) nor entirely opposed (as in adversarial settings), necessitating that each agent balances self-interest with the collective interest. This mixed-motive setting is frequently encountered in real-world applications. For example, in traffic control systems, each agent (e.g., a local intersection controller) may aim to minimize local congestion, which can conflict with global traffic flow optimization if not coordinated. A similar tension happens in sequential social dilemmas (SSDs) such as Cleanup or Harvest [18], where agents must invest in public goods (e.g., cleaning waste or harvesting resources judiciously) that benefit the group but do not yield immediate individual rewards.

However, achieving such a balance in mixed-motive settings is inherently challenging. Excessively selfish behavior by agents can deteriorate collective welfare, which, in turn, negatively impacts each agent's own return—creating a vicious cycle that ultimately harms all participants. Additionally, in some scenarios, certain agents must sacrifice their own returns to improve the collective outcome, potentially leading to unfairness. Conversely, an excessive focus on fairness can hinder learning in tasks that require cooperation. Therefore, it is crucial to enhance collective outcome while ensuring fairness by appropriately balancing individual and collective interests.

39th Conference on Neural Information Processing Systems (NeurIPS 2025).

In mixed-motive settings, many approaches adopt reward restructuring by incorporating intrinsic rewards such as social influence [12], formal contracts [9], gifting [23, 17], and inequity aversion [10]. These methods primarily aim to maximize the collective return by mediating the trade-off between self-interest and collective interests. For example, gifting mechanisms promote cooperation by enabling agents to share a portion of their rewards with others. However, despite their effectiveness in inducing cooperation, such reward restructuring may raise fairness concerns, for example, the gifted reward is intrinsic and not part of the task-defined reward that agents are fundamentally trained to maximize. Consider the Cleanup environment: agents only receive extrinsic rewards for collecting apples, yet apples will only regrow if waste is cleaned. It is often observed that some agents specialize in cleaning waste while others collect apples and subsequently gift a portion of their reward to those who sacrificed their own gain. Although this leads to improved collective performance, the agents engaged in waste cleaning never directly receive task rewards from apple collection. This becomes even worse if the agents are trained with the collective return, since some agents are encouraged to clean the waste all the time. Aside from reward restructuring, an approach has been proposed to align individual and collective objectives by adjusting policy gradients toward stable fixed points of the collective return, while still considering individual interests [20]. However, this method does not adequately consider fairness, as it primarily focuses on stability without explicitly addressing the conflict between individual and collective objectives.

In order to enhance cooperation while ensuring fairness, we propose a fair and conflict-aware gradient adjustment method (FCGrad) that dynamically balances gradients derived from individual and collective objectives by explicitly handling conflicts between them. FCGrad first detects the presence of conflicts, and when conflicts are found, it projects one gradient onto the normal plane of the other—preserving one objective's direction while avoiding interference with the other. Notably, FCGrad prioritizes the gradient associated with the lower objective value. For example, if the individual objective is lower than the collective objective, indicating that the agent is in an unfair situation, we project the individual gradient onto the normal plane of the collective gradient and use the result as the final update. This enables cooperation to be enhanced while maintaining fairness by resolving conflicts. We provide theoretical results showing that, under certain assumptions, the proposed gradient method guarantees monotonic non-decreasing improvement in both collective and individual objectives. We further show that the two objectives converge to the same value, leading to all agents' objectives aligning—thus ensuring individual fairness. In addition, we empirically demonstrate the effectiveness of FCGrad in terms of $\alpha$-fairness [25], which captures both performance and fairness, in the Unfair Coin Game and two sequential social dilemma environments: Cleanup and Harvest.

## 2 Background and Related Works

### 2.1 Partially Observable Stochastic Game

A *Partially Observable Markov Game* (POMG) models multi-agent decision-making under uncertainty [21, 4]. A POMG is defined as a tuple $(N, S, \{A_i\}_{i=1}^N, T, \{O_i\}_{i=1}^N, \{R_i\}_{i=1}^N, \gamma)$, where $N$ is the number of agents, $S$ is the set of states, $A_i$ is the action set of agent $i$, $T : S \times A_1 \times \cdots \times A_N \rightarrow \Delta(S)$ is the transition function, $O_i : S \rightarrow \Delta(\mathcal{O}_i)$ is the observation function, $R_i : S \times A_1 \times \cdots \times A_N \rightarrow \mathbb{R}$ is the reward function for agent $i$, and $\gamma \in [0, 1)$ is the discount factor. Here, depending on the reward structure, a POMG can represent various types of multi-agent settings: *cooperative* settings [13, 14, 15, 16], where all agents share an identical reward function (i.e., $r^1 = \cdots = r^N$); *adversarial* settings [8, 31], where agents have directly opposing objectives, often modeled as zero-sum (i.e., $\sum_{i=1}^N r^i = 0$); or *mixed-motive* settings [24, 17], where agents' rewards are neither fully aligned nor strictly opposed, creating simultaneous incentives for both cooperation and competition.

### 2.2 Mixed-motive Coordination in Multi-Agent RL

We consider mixed-motive settings, where agents' self-interest often conflicts with collective outcomes. Let us define the *collective return* as the average of *individual returns*: $R_{col} = \frac{1}{N} \sum_{i=1}^N R^i(s, a)$, where $R^i(s, a)$ is the individual return of Agent $i$. In the context of gradient-based learning, a conflict occurs when the local and collective return gradients are misaligned, that is, when $\nabla_{\theta_i} \mathbb{E}\left[R^i\right] \cdot \nabla_{\theta_i} \mathbb{E}\left[R_{\text{col}}\right] < 0$, where $\theta_i$ denotes the parameters of Agent $i$'s policy.

To enhance cooperation (i.e. maximize collective reward) while avoiding conflicts, a variety of approaches have been proposed, including inequity aversion [10, 30], social influence [12], reciprocal reward shaping [33], formal contract mechanisms [9], and gifting-based cooperation [23, 17]. Many of these approaches are studied in the context of *Sequential Social Dilemmas (SSDs)* [18], a prominent class of mixed-motive settings in which agents repeatedly arbitrate between short-term selfish actions and long-term collective returns. For example, [17] proposed a gift-based method that balances altrusim and self-interest based based on social relationships between agents. [12] proposed an intrinsic motivation method that rewards agents for exerting causal influence over others' actions, thereby improving coordination in SSDs. [10] introduced inequity-averse agents that learn to cooperate by assigning temporal credit to prosocial behavior and penalizing inequitable outcomes. The aforementioned methods can be broadly viewed as forms of reward shaping, wherein additional intrinsic or socially-informed rewards guide agents toward cooperative behavior.

In contrast to reward shaping approaches, recent work [20] has explored direct optimization in the gradient space to reconcile individual and collective objectives. Specifically, the Altruistic Gradient Adjustment (AgA) method [20] modifies the policy gradients of both the collective and individual objectives, pulling agents toward stable fixed points of the collective objective and pushing them away from unstable ones. The adjusted gradient for Agent $i$ is defined as $g_{aga}^i = g_{col} + \lambda(g_{ind}^i + H_{col}^T g_{col})$, where $g_{col}$ and $g_{ind}$ are the gradients of the collective and individual objectives for Agent $i$, $H_{col}^T$ is the Hessian of the collective return with respect to the policy parameters, and $\lambda$ is the adjustment coefficient and its sign is determined by $\text{sign}[(g_{col} \cdot H_{col}^T g_{col}) \left[(g_{ind}^i \cdot H_{col}^T g_{col}) + \|H_{col}^T \cdot g_{col}\|^2\right]]$. This adjustment steers the update direction according to the local stability of the collective objective. Despite its effectiveness, AgA incurs additional computational complexity, focuses on the stability of the collective objective rather than directly resolving gradient conflicts, and provides no guarantees of monotonic improvement or fairness.

## 2.3 Gradient Adjustment

Gradient adjustment approaches have been actively investigated in multi-task learning [32, 22, 26, 28]. For example, CAGrad [22] formulates a quadratic program to compute a conflict-averse convex combination of gradients, achieving better trade-offs at the cost of increased complexity, and Nash-MTL [26] frames the task-weighting problem as a bargaining game, using the Nash bargaining solution to promote fairness and efficiency across tasks. Another method that inspires our work is PCGrad [32], which mitigates conflicts by projecting each conflicting gradient onto the normal plane of the other, offering a simple yet effective solution with low computational overhead. Specifically, when two gradients $g_1$ and $g_2$ are conflicted, PCGrad adjusts them by projecting one onto the normal plane of the other, i.e., $\tilde{g}_1^{PCGrad} = g_1 - \frac{g_1 \cdot g_2}{\|g_2\|^2} g_2$, and then uses the average of $\tilde{g}_1^{PCGrad}$ and $\tilde{g}_2^{PCGrad}$ as the final update.

## 2.4 Fairness in Multi-agent RL

Fairness concerns how returns are distributed among agents rather than how large the total return is, making it complementary, but often orthogonal to cooperation and efficiency. Fairness has been considered in multi-agent RL literature in both cooperative and mixed-motive settings [34, 6, 1, 29]. For example, in cooperative settings, [34] formulates fairness as the optimization of a fair social welfare function and [6] proposes a method for achieving team fairness by enforcing permutation-equivariant policies, which mitigate emergent unfairness caused by asymmetric role assignment. In mixed-motive settings, [17] shows enhanced fairness when measuring the sum of individual rewards and gifts, whereas in this paper we evaluate fairness using individual rewards only. [10], inspired by the literature on inequality in economics [5], explicitly leverages fairness by adding both disadvantage and advantage inequality terms to the reward of each agent to improve cooperation in SSD. Specifically, the shaped reward for Agent is $r^i = r^i - \alpha_{IA}/(N-1) \sum_{j \neq i} \max(r_j - r_i, 0) - \beta_{IA}/(N-1) \sum_{j \neq i} max(r_i - r_j, 0)$, where $\alpha_{IA}$ and $\beta_{IA}$ weight disadvantage and advantage inequity, respectively.

Note that throughout this paper, we define fairness in terms of task-defined extrinsic individual rewards, the quantities that agents are fundamentally trained to maximize, and do not consider intrinsic rewards such as gifting, as they do not directly reflect actual participation in the underlying task. A more detailed discussion on this assumption is provided in Appendix A.

**Algorithm 1:** FCGrad

---

**Input:** Policy parameters $\theta$, learning rate $\eta$,
weighting factor $\beta$

1 Compute $g_{\text{ind}} := \nabla_\theta V_{\text{ind}}(\theta)$,
$g_{\text{col}} := \nabla_\theta V_{\text{col}}(\theta)$

2 **if** $\langle g_{ind}, g_{col} \rangle \geq 0$ **then**

3     $g_{\text{FCGrad}} \leftarrow (1-\beta)g_{\text{ind}} + \beta g_{\text{col}}$;

4 **else**

5     **if** $V_{col}(\theta) \geq V_{ind}(\theta)$ **then**

6        $g_{\text{FCGrad}} \leftarrow g_{\text{ind}} - \dfrac{\langle g_{\text{col}}, g_{\text{ind}} \rangle}{\|g_{\text{col}}\|^2} g_{\text{col}}$;

7     **else**

8        $g_{\text{FCGrad}} \leftarrow g_{\text{col}} - \dfrac{\langle g_{\text{ind}}, g_{\text{col}} \rangle}{\|g_{\text{ind}}\|^2} g_{\text{ind}}$;

9 **Return** $\theta \leftarrow \theta + \eta g_{\text{FCGrad}}$;

---

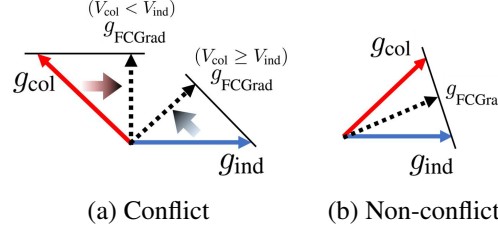

(a) Conflict     (b) Non-conflict

Figure 1: FCGrad illustration: (a) When conflicts occur, the gradient corresponding to the lower objective—either individual or collective—is projected onto the normal plane of the gradient of the higher objective; (b) When no conflict is detected, a task-dependent weighted sum of the two gradients is applied.

## 3 Methodology

In mixed-motive settings, the individual and collective objectives may be either aligned or in conflict. When they are aligned, optimizing both objectives is sufficient, as neither interferes with the other. In such cases, an appropriately weighted combination of the two can be effective. However, when the objectives are in conflict, it becomes essential to explicitly address the interference between them, as prioritizing one may hinder the other. This is because focusing solely on the individual objective may hinder learning in tasks where cooperative behavior is essential for maximizing individual returns, while focusing solely on the collective objective may compromise fairness among agents. Therefore, it is important to (1) recognize when such conflicts arise and (2) correspondingly adjust the individual and collective objectives, appropriately considering both fairness and cooperation.

To this end, we propose a fair and conflict-aware gradient adjustment method, called FCGrad, which guarantees the monotonic non-decrease of both individual and collective objectives, while preserving fairness across individual objectives. Specifically, when the individual and collective gradients are in conflict, FCGrad projects the gradient associated with the lower expected return onto the normal plane of the other. This projected gradient remains a valid ascent direction for its own objective while avoiding interference with the other, and is then used as the final update. For example, if the individual expected return is lower than the collective expected return, indicating that the agent is disadvantaged in terms of fairness, we project the gradient of the individual objective onto the normal plane of the collective gradient and use it as the update direction. The detailed procedure and a visual illustration of FCGrad are provided in Algorithm 1 and Fig. 1, respectively. In the following, we present the detailed method along with its theoretical analysis.

### 3.1 FCGrad: Fair and Conflict-aware Gradient Adjustment

We now describe how FCGrad operates from the perspective of Agent $i$. Let $\theta \in \mathbb{R}^d$ denote the parameters of the policy $\pi_\theta$ for Agent $i$. Let us define $V_{\text{ind}}(\theta)$ and $V_{\text{col}}(\theta)$ as the expected individual and collective returns, respectively, computed under the initial state distribution. Note that $V_{\text{ind}}(\theta)$ and $V_{\text{col}}(\theta)$ are the individual and collective objectives, respectively. Let $g_{\text{ind}} := \nabla_\theta V_{\text{ind}}(\theta)$ and $g_{\text{col}} := \nabla_\theta V_{\text{col}}(\theta)$ denote the gradients of the individual and collective objectives, respectively. These represent ascent directions for $V_{\text{ind}}(\theta)$ and $V_{\text{col}}(\theta)$, meaning that for a sufficiently small $\eta > 0$, the following holds: $V_{\text{ind}}(\theta + \eta g_{\text{ind}}) > V_{\text{ind}}(\theta)$ and $V_{\text{col}}(\theta + \eta g_{\text{col}}) > V_{\text{col}}(\theta)$.

FCGrad proceeds as follows: (1) check whether $g_{\text{ind}}$ and $g_{\text{col}}$ are in conflict by examining the sign of their inner product, where a negative inner product indicates the presence of a conflict. (2) if $\langle g_{\text{ind}}, g_{\text{col}} \rangle \geq 0$ (i.e., non-conflict), FCGrad uses the weighted sum of two gradients: $g = (1-\beta)g_{\text{ind}} + \beta g_{\text{col}}$, (3) $\langle g_{\text{ind}}, g_{\text{col}} \rangle < 0$ (i.e., conflict), FCGrad places more weight on the individual (collective) gradient when the collective (individual) objective is greater, in order to ensure fairness.

The corresponding gradient is given by

$$g_{\text{FCGrad}} = \begin{cases} \tilde{g}_{ind} & \text{if } (V_{\text{col}} \geq V_{\text{ind}}) \\ \tilde{g}_{col} & \text{if } (V_{\text{col}} < V_{\text{ind}}) \end{cases} \tag{1}$$

where $\tilde{g}_{\text{col}}$ and $\tilde{g}_{\text{ind}}$ are the projections of $g_{\text{col}}$ and $g_{\text{ind}}$, respectively, onto the normal plane of another gradient vector, given by

$$\tilde{g}_{\text{col}} := g_{\text{col}} - \frac{\langle g_{\text{ind}}, g_{\text{col}} \rangle}{\|g_{\text{ind}}\|^2} g_{\text{ind}}, \qquad \tilde{g}_{\text{ind}} := g_{\text{ind}} - \frac{\langle g_{\text{col}}, g_{\text{ind}} \rangle}{\|g_{\text{col}}\|^2} g_{\text{col}} \tag{2}$$

(4) update the policy parameter with the step size $\eta$: $\theta \leftarrow \theta + \eta g$. Note that $\tilde{g}_{\text{col}}$ projects $g_{\text{col}}$ onto the normal plane of $g_{\text{ind}}$. Thus, $\tilde{g}_{\text{col}}$ is still a valid ascent direction for the collective objective while preserving the individual reward. This indicates that FCGrad prioritizes the individual objective without compromising the collective one when the agent is in an unfair situation, i.e., when the individual objective is lower. Conversely, when the collective objective is lower, FCGrad prioritizes the collective objective without compromising the individual one.

## 3.2 Theoretical Analysis

In this section, we prove that FCGrad guarantees monotonically non-decreasing improvements in both the collective and individual objectives, and that both objectives converge to the same value. This ensures that the expected individual returns across agents also converge to the same value.

**Theorem 3.1** *Assume $V_{ind}(\theta)$ and $V_{col}(\theta)$ are differentiable and L-smooth. Let the update direction g be defined as in Equation 1. Then, for a sufficiently small step size $\eta > 0$, the update $\theta \leftarrow \theta + \eta g$ yields monotonically non-decreasing improvements in both $V_{col}(\theta)$ and $V_{int}(\theta)$.*

*Proof.* See Appendix B.

Theorem 3.1 states that FCGrad ensures monotonic non-decreasing improvements in both $V_{\text{ind}}(\theta_t)$ and $V_{\text{col}}(\theta_t)$ under certain assumptions. Note that all agents are updated using FCGrad, so both the individual objectives of all agents and the collective objective, defined as the expected return averaged across agents, are improved accordingly. However, monotonic improvement alone does not guarantee fairness. To establish fairness, it is necessary to further show that the individual and collective values converge to the same value over time, which in turn implies that all agents' individual values also become equal. The next theorem formalizes this result by proving that the gap between $V_{\text{ind}}(\theta_t)$ and $V_{\text{col}}(\theta_t)$ vanishes under mild conditions.

**Theorem 3.2** *Assume $V_{ind}(\theta)$ and $V_{col}(\theta)$ be L-smooth, and let $\delta_t := V_{ind}(\theta_t) - V_{col}(\theta_t)$ denote the value gap at iteration t. Assume the step size satisfies the Robbins–Monro conditions: $0 < \eta_t \leq |\delta_t|/L$ with $\sum_t \eta_t = \infty$ and $\sum_t \eta_t^2 < \infty$. Also assume conflict recurrence, meaning that for any $\epsilon > 0$ and any t, if $|\delta_t| \geq \epsilon$, then there exists $t' \geq t$ such that $(g_{ind,t'} \cdot g_{col,t'}) < 0$. Then, the value gap converges to zero:*

$$\lim_{t \to \infty} |V_{ind}(\theta_t) - V_{col}(\theta_t)| = 0. \tag{3}$$

*Proof.* See Appendix B.

Theorem 3.2 states that the gap between the collective and individual objectives converges to zero under certain assumptions, including conflict recurrence, where conflicts occur continuously. This assumption is reasonable in mixed-motive settings, especially near equilibrium, because agents face inherent tensions between cooperation and self-interest, and as they approach equilibrium, misalignments in their objectives can continue to induce conflicts, even with small policy updates. Under the assumption that all agents use FCGrad, the individual objectives of all agents converge to the collective objective, and thus all individual objectives converge to the same value. This, in turn, implies that individual fairness is achieved.

## 3.3 Practical Algorithm

We now introduce a practical FCGrad-based multi-agent RL algorithm for mixed-motive settings. We consider decentralized training and execution, where each agent does not have access to other

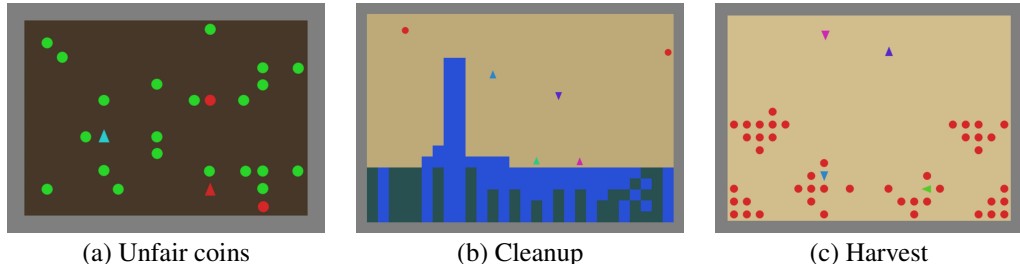

| (a) Unfair coins | (b) Cleanup | (c) Harvest |

Figure 2: The environments considered in our experiments: (a) Unfair Coins—green coins appear more frequently than red coins, inducing fairness challenges; (b) Cleanup with distinct spawn positions—two agents (cyan and pink) spawn near waste areas, while the rest (blue and purple) spawn farther away; and (c) Harvest with distinct spawn positions—two agents (blue and green) spawn near apple (red) regions, while the rest spawn farther away.

agents' information but shares rewards, as commonly assumed in gifting mechanisms [23, 17]. Each agent trains its policy and value functions for both individual and collective returns solely based on its own local observations and the shared rewards. For this, we construct two separate value networks for the individual and collective objectives, while sharing a common encoder between them. Each value function is trained using generalized advantage estimation [27] to compute the corresponding advantage estimates. Using the two value functions, we compute the policy gradients of the individual and collective objectives, denoted as $g_{\text{ind}}$ and $g_{\text{col}}$, respectively, via the PPO policy gradient. These gradients are then combined using FCGrad to determine the final update direction.

## 4 Experimental Results

### 4.1 Experimental Setup

**Environments**  We conduct our experiments using the JAX-based codebase and environments provided by the SocialJAX suite [7]. We modify the existing environments—Coins, Cleanup, and Harvest—to incorporate a fairness perspective. Specifically, since Cleanup and Harvest already involve inherent fairness dilemmas, we introduce only minor changes by assigning distinct respawn positions to the agents. For the Coin Game, which originally focuses on the conflict between individual and collective objectives, we introduce asymmetry in the potential rewards that agents can obtain, creating a disparity in individual incentives. Fig.2 illustrates the considered environments. We provide detailed descriptions in the following sections.

**Metric**  As our goal is to maximize returns while ensuring fairness, both performance and fairness metrics should be jointly considered for evaluation. We use $\alpha$-*fairness* [25] as the evaluation metric, where, given individual returns $(r_1, \cdots, r_N)$, the fairness utility is defined as

$$U_\alpha(r_1, \cdots, r_N) = \begin{cases} \sum_{i=1}^{N} \frac{r_i^{1-\alpha}}{1-\alpha}, & \text{if } \alpha \neq 1, \\ \sum_{i=1}^{N} \log(r_i), & \text{if } \alpha = 1. \end{cases} \tag{4}$$

Notably, the fairness utility recovers several well-known objectives for specific values of $\alpha$: it corresponds to the collective return when $\alpha = 0$, the geometric mean of individual rewards—also known as Nash Social Welfare—when $\alpha = 1$, and the minimum individual reward when $\alpha \to \infty$. Thus, $\alpha = 0$ reflects no consideration of fairness, and as $\alpha$ increases, the evaluation increasingly prioritizes fairness over aggregate performance. In summary, we consider the following three representative instances of $\alpha$-fairness return in our evaluation: (i) average return (**Mean**, $\alpha = 0$), (ii) geometric mean return (**GeoMean**, $\alpha = 1$), and (iii) minimum individual return (**Min**, $\alpha \to \infty$). Note that $\alpha$-*fairness return considers both performance and fairness*, where $\alpha$ determines the trade-off between them. The reported results are averaged over four random seeds.

**Baselines**  We evaluate FCGrad with six baselines: (a) collective reward optimization (Col), (b) individual reward optimization (Ind), (c) inequity aversion reward restructuring (IA) [10], (d) weighted gradient combination of $g_{\text{ind}}$ and $g_{\text{col}}$ (denoted as Weighted), which corresponds to FCGrad without conflict handling, (e) PCGrad [32], and (f) Altruistic Gradient Adjustment (AgA) [20]. Note

that baselines (d)-(f) use the same architecture as FCGrad, where two separate value functions are trained for individual and collective objectives; they differ only in the policy update rule based on $g_{\text{ind}}$ and $g_{\text{col}}$. All methods are implemented on top of the IPPO [3].

**Hyperparameter** We introduce a hyperparameter $\beta$ for FCGrad, which determines the weight between the collective and individual objectives when there is no conflict. $\beta$ plays a particularly important role in tasks that require high-level cooperation. We set $\beta$ to 0.5, 0.7, and 0.8 for the Unfair Coin Game, Cleanup, and Harvest, respectively. The same values of $\beta$ are used for the baseline method, Weighted. Additional hyperparameters for IPPO are provided in Appendix C.

## 4.2 Unfair Coins

The Coins environment [19] consists of two agents (green and red) and two types of coins, each associated with one of the agents. When a coin appears, it is assigned a color with probabilities $p_{\text{green}}$ and $p_{\text{red}}$. An agent receives a reward of $1$ for collecting any coin, regardless of its color. However, collecting a coin of the opposite color imposes a penalty of $-2$ on the other agent, creating a conflict between individual gain and cooperative behavior. In contrast to the original setting [19], where $p_{\text{green}}$ and $p_{\text{red}}$ are both set to 0.5—so that collecting coins matching each agent's color naturally aligns with fairness and also maximizes the collective reward—we consider an unfair variant where $p_{\text{green}} = 15/16$ and $p_{\text{red}} = 1/16$, introducing an inherent asymmetry in coin appearances. Although optimal collective performance still requires agents to collect coins matching their own color, this setup raises a fairness concern: the green agent receives substantially more rewards due to the higher frequency of green coins. To mitigate this imbalance and achieve a fairer outcome, the green agent must occasionally yield coins to the red agent, sacrificing some collective reward in favor of equity.

**Results.** In the Unfair Coin environment, achieving fairness requires the green agent to yield some of its coins to the red agent, thereby reducing its own reward. In other words, there exists a strong trade-off between collective performance and fairness. Therefore, we particularly focus on the performance trend with respect to $\alpha$, as well as the **Min** performance, which places greater emphasis on fairness—the return of the most disadvantaged agent.

Fig. 3 presents the $\alpha$-fairness returns in the unfair coin environment (top) and the individual return of the green and red agents (bottom). The performance of Col, Ind, and AgA is observed to decrease more dramatically as $\alpha$ increases compared to FCGrad and PCGrad, which are conflict-aware methods. Interestingly, both the collective and individual approaches result in extremely unfair outcomes, but in opposite directions. Col, which maximizes collective reward, trains both agents to collect their own coins. As a result, the green agent, with more coin opportunities, gains higher returns, leading to unfairness.

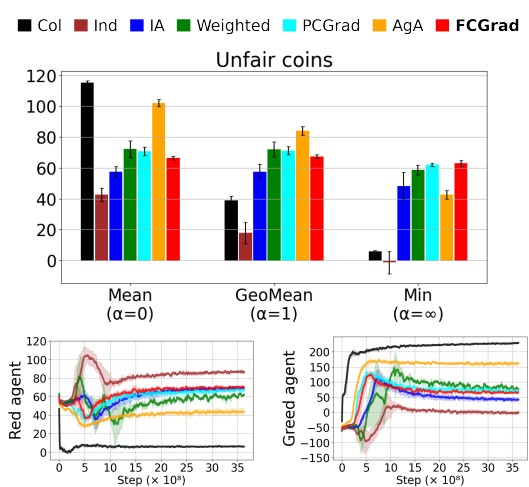

Figure 3: Top: Performance across agents in terms of mean, geometric mean, and minimum. Note that a higher value of $\alpha$ places more emphasis on fairness. Bottom: Agent-wise returns—Red and Green Agents.

In contrast, with the individual objective, the red agent outperforms the green agent, possibly because the green agent is more frequently penalized by negative rewards due to the abundance of green coins. Meanwhile, the red agent learns without such penalties, accelerating its progress. However, FCGrad shows little variation across agents as $\alpha$ changes, indicating achieved fairness. Notably, FCGrad outperforms the baselines in terms of **Min** performance. As shown in Fig. 3, both the green and red agents converge to nearly identical returns, showing that fairness is effectively achieved.

## 4.3 Cleanup

The Cleanup environment consists of $N = 4$ agents, apples, and waste. Each agent receives a reward of $1$ for collecting an apple. Apples grow in an orchard, but their growth depends on the

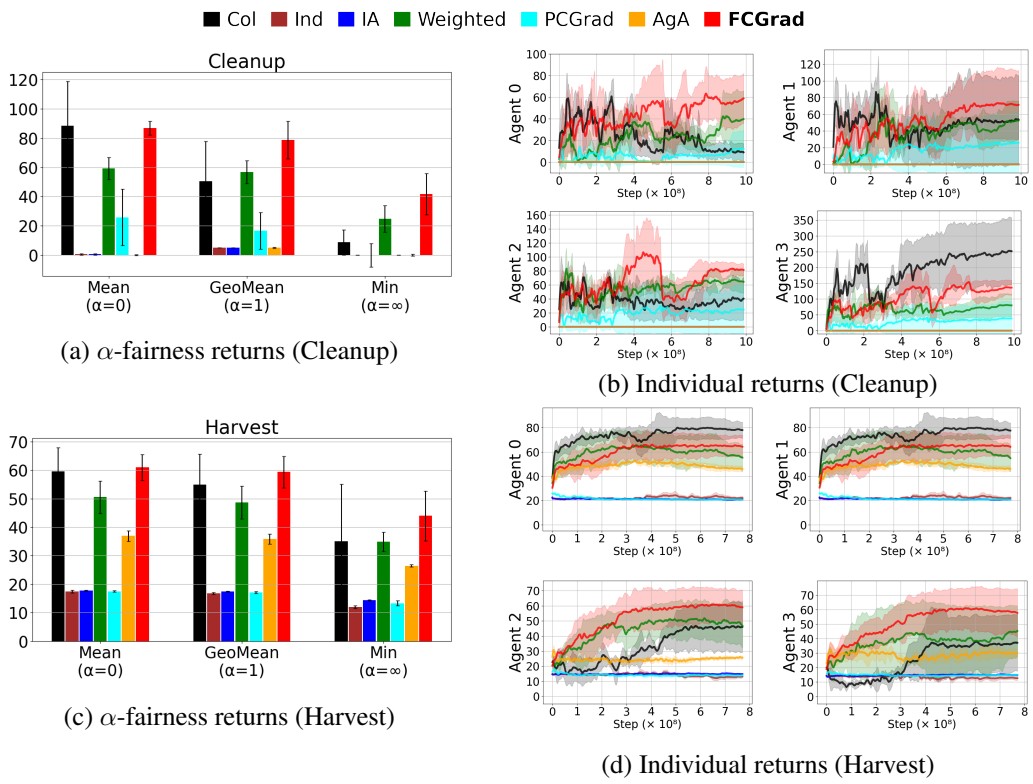

(a) $\alpha$-fairness returns (Cleanup)

(b) Individual returns (Cleanup)

(c) $\alpha$-fairness returns (Harvest)

(d) Individual returns (Harvest)

Figure 4: $\alpha$-fairness returns and individual returns in the cleanup and harvest environments.

amount of waste present in the environment. Waste accumulates at a constant rate, and beyond a certain threshold, apple growth ceases entirely. Therefore, in order to sustain apple regrowth, some agents must sacrifice their immediate reward by cleaning up the waste. This creates a social dilemma, as the necessary act of cleaning benefits the group but does not provide direct individual reward, thereby generating a tension between self-interest and cooperative behavior. In contrast to the original configuration, where agents are randomly spawned across the map, we fix the spawn positions of agents: some (Agents 2 and 3 in our case) are placed near the apple orchard, while others (Agents 0 and 1) are positioned closer to the waste area. This spatial asymmetry further amplifies the conflict between fairness and efficiency. Note that, unlike the Unfair Coin, Cleanup introduces an intertemporal perspective, involving a trade-off between short-term individual interest and long-term collective interest [10].

**Results.** Fig. 4 (a) and (b) show the $\alpha$-fairness performance and individual rewards during training in the Cleanup environment. FCGrad outperforms the baselines in terms of both **GeoMean** and **Min**, which reflect not only total return but also fairness. In addition, FCGrad achieves comparable performance to Col in terms of **Mean**, which is the optimization target of Col. As shown in Fig. 4 (b), under Col, Agent 3 learns to monopolize apple collection, while Agent 0 is trained to sacrifice by primarily cleaning waste. In contrast, FCGrad leads all four agents to obtain reasonably similar returns—demonstrating more fair behavior and achieving the best result in terms of **Min**. Since using the collective reward is essential in this environment, methods that rely heavily on individual rewards, such as Ind and IA, fail to learn effectively. In addition, AgA fails to properly balance between individual and collective objectives, also struggle to learn successfully.

## 4.4 Harvest

The Harvest environment features $N = 4$ agents and apples distributed across orchard patches. Each agent receives a reward of 1 per apple, but regrowth is stochastic and depends on nearby apples within a fixed radius. Over-harvesting depletes resources, risking environmental collapse, and thus agents must coordinate implicitly to sustain long-term returns. This creates a social dilemma between

short-term individual gain and long-term collective benefit. We also introduce spatial asymmetry: Agents 0 and 1 spawn near apples, while Agents 2 and 3 spawn farther away, making collection easier for the former. Similar to the Cleanup, Harvest also poses intertemporal challenges for both cooperation and fairness.

**Results.** Fig. 4 (c) and (d) show the $\alpha$-fairness returns and individual agent returns during training. FCGrad outperforms the baselines across the considered $\alpha$ values. With the Col, Agents 0 and 1 achieve higher returns than Agents 2 and 3, indicating that they focus solely on collecting apples while accounting for the intertemporal dilemma, but not addressing the resulting unfairness toward Agents 2 and 3. In contrast, FCGrad leads all four agents to achieve similar returns, implying that Agents 0 and 1 take into account the outcomes of Agents 2 and 3. Similar to the results in Cleanup, methods that rely heavily on individual rewards, such as Ind and IA, perform poorly, though they achieve marginal learning. AgA performs better than the individual-reward-based methods, but still underperforms compared to FCGrad.

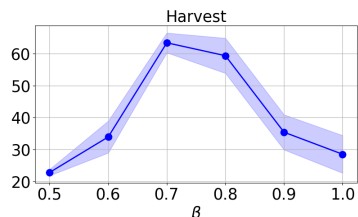

Figure 5: GeoMean of FCGrad with respect to $\beta$ in the harvest environment.

## 4.5 Additional Analysis: Ablation and Fairness Metrics

**Weighting factor:** $\beta$ determines the balance between the collective and individual objectives when no conflict is detected. It plays a particularly important role in tasks that require high-level cooperation. For example, in Cleanup, ignoring the collective objective makes it difficult for agents to discover how to improve their individual rewards. We observed this phenomenon in the previous section—solely maximizing individual rewards does not perform well. We present the **GeoMean** performance of FCGrad in the Harvest environment in Fig. 5, which shows that a $\beta$ value between 0.7 and 0.8 yields the best performance. Thus, $\beta$ reflects the required degree of cooperation over self-interest.

| | Coin | | Cleanup | | Harvest | |
|---|---|---|---|---|---|---|
| **Alg** | Gini | Jain | Gini | Jain | Gini | Jain |
| Col | 0.474 | 0.526 | 0.558 | 0.432 | 0.182 | 0.882 |
| Ind | 0.509 | 0.498 | 0.522 | 0.515 | 0.136 | 0.936 |
| IA | 0.122 | 0.942 | 0.536 | 0.497 | **0.087** | **0.973** |
| Weighted | 0.048 | 0.991 | **0.266** | **0.801** | 0.146 | 0.928 |
| PCGrad | **0.039** | **0.994** | 0.469 | 0.572 | 0.101 | **0.965** |
| AgA | 0.238 | 0.749 | 0.331 | 0.723 | 0.123 | 0.948 |
| FCGrad | **0.010** | **0.999** | **0.223** | **0.835** | **0.093** | 0.959 |

Table 1: Addtional fairness evaluation using Gini coefficient and Jain's index. Lower Gini and higher Jain indicate greater fairness. Top-2 most fair scores in each column are highlighted in bold.

**Additional Fairness metrics:** We additionally evaluate fairness using the Gini coefficient [2] and Jain's index [11]. The Gini coefficient is defined as $\text{Gini}(r_1, \cdots, r_N) = \frac{\sum_{i=1}^{N} \sum_{j=1}^{N} |r_i - r_j|}{2N \sum_{i=1}^{N} r_i}$ and Jain's index is defined as $\text{Jain}(r_1, \cdots, r_N) = \frac{\left(\sum_{i=1}^{N} r_i\right)^2}{N \sum_{i=1}^{N} r_i^2}$, where both metrics range between 0 and 1 and lower Gini and higher Jain values indicate better fairness. Table 1 presents the results, showing that FCGrad generally achieves superior fairness.

## 5 Conclusion

In this work, we address the long-standing challenge of achieving both cooperation and fairness in mixed-motive multi-agent RL. We propose FCGrad, a conflict-aware gradient adjustment method that explicitly resolves gradient-level conflicts between individual and collective objectives. FCGrad dynamically adjusts the update direction based on which objective is more disadvantaged by projecting one gradient onto the normal plane of the other. We theoretically prove that this mechanism guarantees monotonic improvement and convergence of both objectives to the same value. Consequently, individual objectives across agents also converge, ensuring fairness. Extensive experiments in the Unfair Coin environment and sequential social dilemma settings, Cleanup and Harvest, demonstrate that FCGrad not only improves overall performance but also achieves superior fairness, as measured by $\alpha$-fairness return metrics.

**Limitation** In practice, the recurrence of gradient conflicts, required for our theoretical guarantee, may not hold, as it can be influenced by the weighting factor in non-conflict cases. Understanding this interplay is a promising direction for future work.

**Broader Impact** Our work promotes fairness in learned behaviors, potentially preventing emergent inequalities in decentralized systems. We believe it has a positive societal impact.

## 6 Acknowledgement

This work was supported by the ONR MURI grant N00014-25-1-2116.

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

# A  Discussion on Fairness

In this appendix, we clarify the modeling assumption underlying our notion of fairness and contrast it with alternative perspectives such as reward-redistribution-based fairness. Our formulation intentionally focuses on a different regime: fairness is grounded purely in task-defined extrinsic rewards that directly reflect each agent's actual behavior, rather than assuming the availability of contract or currency-like mechanisms (e.g., reward exchanges or gifting) for compensating agents.

## A.1  Extrinsic-Reward-Based Fairness vs. Reward Redistribution

Prior approaches such as gifting or incentive mechanisms (e.g., [9, 17, 23]) allow agents to redistribute rewards among one another—often interpreted as a currency-like signal or contract that enables division of labor. Under such assumptions, reward transfers are considered real, tangible returns and can be used to compensate agents for sacrificial roles (e.g., pollution cleaning without harvesting apples).

In contrast, our work adopts a more *primitive* perspective of fairness: we consider only *task-defined extrinsic rewards* that arise directly from environment state–action outcomes (e.g., rewards from collecting apples in the Cleanup environment). We intentionally do not assume the existence of an auxiliary payment mechanism, such as money or transferable reward tokens, that is external to the environment dynamics. From this viewpoint, if one agent continuously cleans while others only harvest apples, such an outcome is deemed unfair unless the cleaner also receives direct extrinsic returns. This modeling choice focuses on fairness that reflects *actual participation in the task*, rather than contractual compensation.

## A.2  Pareto Optimality and the Role of $\alpha$-Fairness

We emphasize that our objective is not to compute or approximate a game-theoretic equilibrium (e.g., Nash or correlated equilibrium), but rather to learn *Pareto-optimal* outcomes. In particular, $\alpha$-fairness is used only as an evaluation metric, not as a training objective. By varying $\alpha$, one can evaluate different trade-offs between pure efficiency ($\alpha = 0$), multiplicative balance ($\alpha = 1$), and max-min fairness ($\alpha \to \infty$). FCGrad yields outcomes that lie on the Pareto frontier across these trade-offs: in terms of collective return it performs comparably to the best baselines, while in terms of $\alpha = 1$ or $\alpha = \infty$ it significantly improves fairness without degrading performance.

# B    Theoretical Results

**Lemma B.1** *Let $J : \mathbb{R}^d \to \mathbb{R}$ be a continuously differentiable and L-smooth function. Let $g_1 = \nabla_\theta J(\theta)$ be the gradient of J at point $\theta$, and let $g_2 \in \mathbb{R}^d$ be any vector satisfying $\langle g_1, g_2 \rangle > 0$. Then, for small step size $\eta < \frac{2\langle g_1, g_2 \rangle}{L\|g_2\|^2}$, the update $\theta \leftarrow \theta + \eta g_2$ yields a strict improvement:*

$$J(\theta + \eta g_2) > J(\theta).$$

*Proof.* Since $J$ is $L$-smooth, for any $\theta \in \mathbb{R}^d$, update direction $g_2 \in \mathbb{R}^d$, and step size $\eta > 0$, the following inequality holds:

$$J(\theta + \eta g_2) \geq J(\theta) + \eta \langle \nabla_\theta J(\theta), g_2 \rangle - \frac{L}{2}\eta^2 \|g_2\|^2.$$

Let $g_1 = \nabla_\theta J(\theta)$. Then:

$$J(\theta + \eta g_2) \geq J(\theta) + \eta \langle g_1, g_2 \rangle - \frac{L}{2}\eta^2 \|g_2\|^2.$$

Define the right-hand side as a function of $\eta$:

$$\Delta(\eta) := \eta \langle g_1, g_2 \rangle - \frac{L}{2}\eta^2 \|g_2\|^2.$$

Since $\langle g_1, g_2 \rangle > 0$, this is a concave quadratic function that is positive for small enough $\eta$. Specifically, the inequality $\Delta(\eta) > 0$ holds when:

$$\eta < \frac{2\langle g_1, g_2 \rangle}{L\|g_2\|^2}.$$

Therefore, for any $\eta \in \left(0, \frac{2\langle g_1, g_2 \rangle}{L\|g_2\|^2}\right)$, we have:

$$J(\theta + \eta g_2) > J(\theta).$$

**Theorem B.2** *Assume $V_{ind}(\theta)$ and $V_{col}(\theta)$ are differentiable and L-smooth. Let the update direction $g$ be defined as in Equation 1. Then, for a sufficiently small step size $\eta > 0$, the update $\theta \leftarrow \theta + \eta g$ yields monotonically non-decreasing improvements in both $V_{col}(\theta)$ and $V_{int}(\theta)$.*

We consider three cases:

**Case 1:** (Non-conflict) $g_{ind} \cdot g_{col} \geq 0$. Then $g = \beta g_{ind} + (1 - \beta)g_{col}$. Since $g_{ind}, g_{col}$ are ascent directions for $V_{ind}, V_{col}$, respectively, their convex combination also satisfies:

$$g_{ind} \cdot g = \beta \|g_{ind}\|^2 + (1 - \beta)g_{ind} \cdot g_{col} > 0 \tag{5}$$

$$g_{col} \cdot g = \beta g_{col} \cdot g_{ind} + (1 - \beta)\|g_{col}\|^2 > 0 \tag{6}$$

Since $g_{ind} \cdot g$ and $g_{ind} \cdot g$ are positive, according to Lemma 3.1, $g$ yields a strict improvement in both $V_{ind}$ and $V_{col}$.

**Case 2:** (Conflict) $g_{ind} \cdot g_{col} < 0$ and $V_{ind}(\theta) < V_{col}(\theta)$. We then use: $g = g_{ind} - \frac{g_{col} \cdot g_{ind}}{\|g_{col}\|^2} g_{col}$. Now,

$$g_{ind} \cdot g = g_{ind} \cdot g_{ind} - \frac{(g_{ind} \cdot g_{col})}{\|g_{col}\|^2}(g_{ind} \cdot g_{col}) = \frac{\|g_{ind}\|^2\|g_{col}\|^2 - (g_{ind} \cdot g_{col})^2}{\|g_{col}\|^2} > 0$$

$$g_{col} \cdot g = g_{col} \cdot g_{ind} - \frac{(g_{ind} \cdot g_{col})}{\|g_{col}\|^2}\langle g_{col}, g_{col} \rangle = 0 \tag{7}$$

Since $g_{ind} \cdot g$ is positive, according to Lemma 3.1, $g$ yields a strict improvement in $V_{ind}$. In addition, since $g_{col} \cdot g$ is zero, $g$ does not decrease $V_{col}$.

**Case 3:** (Conflict) $g_{ind} \cdot g_{col} < 0$ and $V_{ind}(\theta) > V_{col}(\theta)$. Symmetric to Case 2: $g$ yields a strict improvement in both $V_{col}$ and does not decrease $V_{ind}$.

Thus, in all cases, $g$ induces monotonically non-decreasing improvements in $V_{ind}$ and $V_{col}$.

**Lemma B.3 (Single conflict step)** *When the conflict happens (i.e., $(g_{ind} \cdot g_{col}) < 0$), then for sufficiently small step size, $0 \le \eta_t \le \|\delta_t\|/L$, we have*

$$L_{t+1} - L_t \le -\frac{\eta_t}{2}\,|\delta_t|\,\|d_t\|^2. \tag{8}$$

*Proof.* When $\delta_t < 0$ (i.e. $V_{col} > V_{ind}$), we use $g = g_{ind} - \frac{g_{ind} \cdot g_{col}}{\|g_{col}\|^2} g_{col}$. Since $L$ is L-smooth function, the following holds

$$L_{t+1} - L_t \le \eta_t(\nabla L_t \cdot g) + \frac{L}{2}\eta_t^2\|g\|^2 \tag{9}$$

Here,

$$(\nabla L_t \cdot g) = \delta_t(g_{ind} - g_{col}) \cdot g = \delta_t(g_{ind} \cdot g - g_{col} \cdot g) = \delta_t(g_{ind} \cdot g) = \delta_t\|g\|^2 \tag{10}$$

Thus, we have

$$L_{t+1} - L_t \le \eta_t\delta_t\|g\|^2 + \frac{L}{2}\eta_t^2\|g\|^2\| \le \eta_t\delta_t\|g\|^2 + \frac{\|\delta_t\|\eta_t}{2}\|g\|^2 = -\frac{\eta_t}{2}\|\delta_t\|\|g\|^2 \tag{11}$$

**Lemma B.4 (Single non-conflict step)** *When the conflict does not happen, (i.e., $(g_{ind} \cdot g_{col}) \ge 0$), the proposed gradient is used. We assume that the step size meets the Robbins-Monro conditions (i.e. $\sum_{t=0}^{\infty} \eta_t = \infty$, $\sum_{t=0}^{\infty} \eta_t^2 < \infty$.) Then, the following holds:*

$$\sum_{t \in \mathcal{N}} \|L_{t+1} - L_t\| < \infty \tag{12}$$

*where $\mathcal{N}$ is the set of all non-conflict indices.*

*Proof.* $g = \beta g_{ind} + (1 - \beta)g_{col}$. Let us define $G := \sup_t\big(\|g_{1,t}\| + \|g_{2,t}\|\big)\ (< \infty)$.

Since $L$ is L-smooth, we have

$$L_{t+1} - L_t \le \eta_t\langle\nabla_\theta L_t,\, g\rangle + \frac{L}{2}\,\eta_t^2\|g\|^2. \tag{13}$$

Since $\nabla_\theta L_t = \delta_t(g_{ind} - g_{col})$,

$$\|\langle\nabla_\theta L_t,\, g\rangle\| = \|\delta_t\langle g_{ind} - g_{col}, \beta g_{ind} + (1 - \beta)g_{col}\rangle\| \tag{14}$$

$$\le |\delta_t|\Big[\beta\|g_{ind}\|\|g_{ind} - g_{col}\| + (1 - \beta)\|g_{col}\|\|g_{ind} - g_{col}\|\Big] \quad \text{(Cauchy-Schwarz)} \tag{15}$$

$$\le |\delta_t|\Big[\beta\|g_{ind}\| + (1 - \beta)\|g_{col}\|\Big]2G \le 2G^2|\delta_t|. \tag{16}$$

Based on the assumption of the step size $\eta$ ($\eta_t \le |\delta_t|/L$), we have

$$|\eta_t\langle\nabla_\theta L_t, d_t\rangle| \le 2G^2\,|\delta_t|\,\eta_t \le 2G^2 L\,\eta_t^2. \tag{C}$$

Since $\|g\| = \|\beta g_{ind} + g_{col}\| \le \beta\|g_{ind}\| + (1 - \beta)\|g_{col}\| \le G$, the following holds.

$$\frac{L}{2}\eta_t^2\|g\|^2 \le \frac{L}{2}\eta_t^2 G^2 \tag{17}$$

Combined above, we have

$$\|L_{t+1} - L_t\| \le (2G^2 L + \frac{L}{2}G^2)\eta_t^2 = \frac{5}{2}G^2 L\eta_t^2 \tag{18}$$

Define $C_0 := \frac{5}{2}\,G^2 L$ to obtain

$$|L_{t+1} - L_t| \le C_0\,\eta_t^2.$$

Because $\sum_{t=0}^{\infty} \eta_t^2 < \infty$ (Robbins–Monro assumption),

$$\sum_{t \in \mathcal{N}} |L_{t+1} - L_t| \le C_0 \sum_{t \in \mathcal{N}} \eta_t^2 \le C_0 \sum_{t=0}^{\infty} \eta_t^2 < \infty.$$

**Theorem B.5** *Let $V_{ind}$ and $V_{col}$ be L-smooth. Assume the step size satisfies the Robbins–Monro conditions: $0 < \eta_t \leq |\delta_t|/L$ with $\sum_t \eta_t = \infty$ and $\sum_t \eta_t^2 < \infty$. Also assume conflict recurrence, meaning that for any $\epsilon > 0$ and any $t$, if $|\delta_t| \geq \epsilon$, then there exists $t' \geq t$ such that $(g_{ind,t'} \cdot g_{col,t'}) < 0$. Then, the value gap converges to zero:*

$$\lim_{t \to \infty} |V_{ind}(\theta_t) - V_{col}(\theta_t)| = 0. \tag{19}$$

*Proof.* Denote conflict indices by $\mathcal{C}$ and non-conflict by $\mathcal{N}$. Lemma A.3 and Lemma A.4 give for every horizon $T$

$$L_T \ \leq \ L_0 - \frac{1}{2} \sum_{t \in \mathcal{C}, \, t < T} \eta_t \, |\delta_t| \, \|d_t\|^2 + \ C_0 \sum_{t \in \mathcal{N}, \, t < T} \eta_t^2. \tag{20}$$

According to the assumption of the Robbins-Monro, the following holds:

$$\sum_{t \in \mathcal{C}} \eta_t \, |\delta_t| \, \|d_t\|^2 < \infty. \tag{21}$$

For any conflict step the projection property and bounded gradients imply $\|g_t\| \geq \sigma > 0$ with $\sigma := \frac{1}{2} \min(\|g_{\text{ind},t}\|, \|g_{\text{col},t}\|)$. Thus, we have

$$\sum_{t \in \mathcal{C}} \eta_t \, |\delta_t| \leq \sigma^{-2} \sum_{t \in \mathcal{C}} \eta_t |\delta_t| \|g_t\|^2 < \infty. \tag{22}$$

Here, we use contradiction. Assume $\limsup_{t \to \infty} |\delta_t| = \varepsilon_0 > 0$. Set $\varepsilon := \varepsilon_0/2$. By the assumption, there exists an *infinite* set $\mathcal{C}_\varepsilon = \{ t \in \mathcal{C} \mid |\delta_t| \geq \varepsilon \}$. Then for every $t \in \mathcal{C}_\varepsilon$, $\eta_t |\delta_t| \geq \eta_t \varepsilon$. Because $\sum_t \eta_t = \infty$, $\sum_{t \in \mathcal{C}_\varepsilon} \eta_t \varepsilon = \infty$, contradicting the finiteness of Eq. 22. Therefore, $\limsup_{t \to \infty} |\delta_t| = 0$.

# C   Implementation Details

All experiments were run on a local server equipped with an AMD EPYC 7713 64-Core CPU and five NVIDIA RTX 6000 Ada Generation GPUs. Each rollout consisted of 64–256 parallel environments depending on the task, and training time per run ranged from 2 to 8 hours. The official implementation of FCGrad is available at: `https://github.com/wjkim1202/fcgrad`.

## C.1   Unfair Coin

Each agent has a CNN-based actor-critic network. The observation is processed through three convolutional layers with kernel sizes of $5 \times 5$, $3 \times 3$, and $3 \times 3$, each with 32 channels and ReLU activations, followed by a fully connected layer with 64 units. The actor head outputs a categorical distribution over discrete actions, while the critic consists of two separate heads estimating the individual and collective value functions.

We train the networks using the Adam optimizer with a learning rate of $1 \times 10^{-4}$, linearly annealed over time. PPO is used with a clipping threshold of 0.2 and two update epochs per iteration, using 500 minibatches. We collect trajectories from 256 parallel environments, each running for 1000 steps per rollout. The discount factor is set to $\gamma = 0.99$ and the GAE parameter to $\lambda = 0.95$. The entropy and value loss coefficients are set to 0.1, respectively. Gradients are clipped to a maximum global norm of 0.5.

## C.2   Cleanup

Each agent is equipped with a convolutional actor-critical network. The observation is processed through three convolutional layers with kernel sizes of $5 \times 5$, $3 \times 3$, and $3 \times 3$, each with 32 channels and ReLU activations, followed by a fully connected layer with 64 units. The actor outputs a categorical distribution over discrete actions, and the critic consists of two heads that estimate the individual and collective value functions, respectively.

Training is performed using PPO with a clipping threshold of 0.2 and two update epochs per iteration. A total of 500 minibatches are used per update, with data collected from 64 parallel environments running 1000 steps per rollout. The discount factor is set to $\gamma = 0.99$, and the GAE parameter is set to $\lambda = 0.95$. We use the Adam optimizer with an initial learning rate of $5 \times 10^{-4}$, which is linearly annealed during training. The value loss coefficient and entropy coefficient are both set to 0.01, and the value function loss is weighted by 0.5. Gradients are clipped with a maximum global norm of 0.5.

## C.3   Harvest

Each agent is equipped with a convolutional actor-critical network. The observation is processed through three convolutional layers with kernel sizes of $5 \times 5$, $3 \times 3$, and $3 \times 3$, each with 32 channels and ReLU activations, followed by a fully connected layer with 64 units. The actor outputs a categorical distribution over discrete actions, and the critic consists of two heads that estimate the individual and collective value functions, respectively.

Training is performed using PPO with a clipping threshold of 0.2 and two update epochs per iteration. A total of 500 minibatches are used per update, with data collected from 64 parallel environments running 1000 steps per rollout. The discount factor is set to $\gamma = 0.99$, and the GAE parameter is set to $\lambda = 0.95$. We use the Adam optimizer with an initial learning rate of $5 \times 10^{-4}$, which is linearly annealed during training. The entropy and value function loss coefficients are set to 0.01 and 0.5, respectively. Gradients are clipped to a maximum global norm of 0.5.

