# OpenReview forum: "Fair Cooperation in Mixed-Motive Games via Conflict-Aware Gradient Adjustment"
_NeurIPS.cc/2025/Conference — NeurIPS 2025 spotlight_

### Official Review · Reviewer_ATyi · 2025-06-30

**Clarity:** 3
**Significance:** 3
**Originality:** 3
**Rating:** 4
**Confidence:** 3

**Summary:**

This paper proposes a conflict-aware gradient adjustment method (FCGrad) for multi-agent reinforcement learning that promotes cooperation in mixed-motive games while ensuring fairness among agents.

**Questions:**

These two works seem to have somewhat similar insights. I’m curious if there is any connection between them and would like to hear the author’s thoughts.
- Duque, Juan Agustin, et al. "Advantage Alignment Algorithms." _arXiv preprint arXiv:2406.14662_ (2024).

**Ethical Concerns:**

["NO or VERY MINOR ethics concerns only"]

**Final Justification:**

All issues are resolved.

**Limitations:**

Mentioned in weaknesses.

**Paper Formatting Concerns:**

Should the first letter of ‘Motive’ be capitalized in the title and section headings?

**Quality:**

3

**Strengths And Weaknesses:**

Strengths

- The research problem concerns mixed-motive tasks, which is an issue that the MARL community should pay attention to.
- The method is straightforward and aligns well with intuition.
- The choice of experimental environments is consistent with the MARL setting and supports the authors’ claims.


Weaknesses

- The research focuses on mixed-motive tasks but lacks discussion of solution concepts from game theory. What equilibrium is being targeted here? How does changing the gradient direction help find a better equilibrium?
- Based on the authors’ narrative, it seems to be about the direction of motivation/preference, but the method modifies the gradient direction instead. Is there a gap between these two?
- Fairness and resolving conflict seem to be two separate issues. If I understand correctly, fairness here refers to taking the mean direction in the conflict-free case in Figure 1. But wouldn’t this affect the optimality of the problem?

- The motivation might not be very clear. How is the trade-off between optimality and fairness handled? For example, in the authors’ Cleanup environment, the optimal solution is division of labor — one agent continuously cleans the pollution while the other continuously collects apples, which maximizes efficiency since they don’t switch roles. However, this is unfair. Some prior work adjusts the reward allocation to address this, but here the authors do not introduce any additional mechanism. Since fairness is emphasized, it should affect optimality, so how should the metric be defined in this case?

Overall, I think these are key questions. If there is a reasonable response, I am open to raising the score.

---

> ### Author Rebuttal · Authors · 2025-07-31
>
> We appreciate the reviewer’s valuable comments and questions.
>
> Below, we provide our responses to the reviewer’s concerns and questions.
>
> ## [4-1] Regarding "discussion of solution concepts from game theory"
>
> Regarding "What equilibrium is being targeted here?"
> We clarify that our work does not aim to explicitly compute or characterize a game-theoretic equilibrium (e.g., Nash or correlated equilibrium). Instead, our focus is on learning Pareto-optimal joint policies in mixed-motive multi-agent environments.
>
> In these settings, pure equilibrium solutions (such as Nash equilibrium) often result in inefficient or unfair outcomes. Rather than relying on equilibrium concepts, we directly optimize the joint training dynamics such that all agents’ individual value functions converge, while maximizing the performance. This guarantees that the learned outcome lies on the Pareto front.
>
>
> To evaluate the quality of these outcomes, we adopt the α-fairness framework, which allows us to capture different trade-offs between efficiency and fairness:
> - $\alpha=0$-fairness (equivalent to average return) captures only task completion;
> - $\alpha=1$-fairness (geometric mean) and $\alpha=\infty$-fairness (minimum individual return) incorporate both task performance and fairness, with increasing emphasis on fairness.
>
> We will clarify in the final version that our solution concept is Pareto optimality.
>
>
> ---
>
> ## [4-2] Regarding "the direction of motivation"
>
> We represent the direction of motivation through gradient adjustment. In reinforcement learning, the policy is guided by the value function. FCGrad modifies this guidance by adjusting the gradient direction to promote fairness and cooperation.
>
> ---
>
> ## [4-3] Regarding "Fairness and resolving conflict seem to be two separate issues"
>
> FCGrad aims to achieve fairness through conflict-aware gradient adjustment. Fairness in our context refers to the actual returns received by each agent, not to taking the mean of gradient directions. FCGrad allows agents to prioritize the collective objective when they are in an unfair situation (i.e., receiving less return than others), and vice versa. This adaptive adjustment promotes fairness while still optimizing expected returns. As shown in our theoretical results, this procedure yields a monotonic non-decreasing improvement in the value functions.
>
> Regarding optimality, we emphasize that it depends on the chosen metric. This is precisely why we adopt the $\alpha$-fairness framework, which jointly considers both task completion and fairness. For example, in terms of average return ($\alpha=0$), which is widely used in the literature and ignores fairness, FCGrad performs comparably to the Col baseline (which maximizes collective return) in the Cleanup and Harvest environments, and even slightly worse in Unfair Coin. However, in terms of $\alpha=1$ and $\alpha=\infty$, which incorporate fairness to increasing degrees, FCGrad consistently outperforms the baselines. This demonstrates that FCGrad achieves a favorable trade-off between fairness and performance.
>
> More broadly, $\alpha$-fairness defines a family of Pareto-optimal solutions with varying emphasis on fairness. By adjusting gradients to navigate toward different regions of the Pareto front, FCGrad enables a flexible trade-off between fairness and efficiency, rather than sacrificing one for the other.
>
>
> ---
>
> ## [4-4] Regarding "motivation"
>
> The example provided by the reviewer—division of labor that maximizes average return but is unfair—is precisely the kind of solution FCGrad is designed to address (as also mentioned in the Introduction). As shown in Figure 4(a), the Col baseline, which optimizes collective return, performs poorly in terms of Min ($\alpha = \infty$), indicating that some agents receive disproportionately low rewards under "efficient but unfair" policies. In contrast, FCGrad achieves a minimum return close to 40, suggesting that even the least-rewarded agent is actively collecting apples.
>
> While some prior works attempt to address fairness through reward shaping (as discussed in Section 2.4), these typically rely on gifting mechanisms—where agents transfer rewards to others. Fairness is then measured over the sum of task rewards and gifts. However, this can be problematic: for example, one agent may clean pollution alone while receiving only "intrinsic" rewards, which does not reflect actual participation in the task. In contrast, we define fairness solely based on task rewards to better capture each agent’s true contribution.
>
> FCGrad addresses this imbalance through gradient adjustment. Specifically, when an agent’s individual return is lower than the collective return, FCGrad updates its policy in the direction of its individual value function (e.g., toward collecting apples). Conversely, agents whose individual return exceeds the collective return shift their gradient toward the collective objective (e.g., cleaning pollution). This dynamic leads to more balanced task participation and promotes fairness while maintaining high overall performance.
>
> To capture the trade-off between fairness and optimality, we adopt $\alpha$-fairness as a unified metric. For $\alpha > 0$, it balances both aspects, with higher $\alpha$ placing more emphasis on fairness. As our results show, FCGrad outperforms baselines in terms of $\alpha = 1$ and $\alpha = \infty$, achieving strong fairness while maintaining high task performance. To the best of our knowledge, we are the first to report $\alpha$-fairness as an evaluation metric in mixed-motive multi-agent reinforcement learning.
>
> ---
>
> ## 4-5) Regarding "connection with [Duque et al.]"
>
> We appreciate the reviewer pointing out the work by Duque et al., which indeed shares a high-level motivation with ours—namely, addressing the misalignment between individual and collective objectives, particularly in social dilemmas where selfish behavior leads to suboptimal outcomes.
>
> While both works aim to promote cooperation, their approaches differ fundamentally. Duque et al. attempt to resolve this misalignment by aligning agents' advantages, shaping the behavior of other agents to promote robust cooperation. Their method seeks to mitigate the conflict, rather than leveraging it.
>
> In contrast, our method explicitly detects and addresses such conflicts during training via conflict-aware gradient adjustments. Moreover, fairness is a core objective in our framework. Unlike Duque et al., which does not incorporate fairness metrics, FCGrad is designed to promote both cooperation and fairness by dynamically adjusting the learning direction based on disparities in agent returns.
>
> In summary, while the high-level motivation is similar, our objectives and mechanisms differ: Duque et al. focus on advantage alignment to foster cooperation, whereas we explicitly exploit gradient-level conflicts to achieve fair and cooperative outcomes.

---

> > ### Comment · Reviewer_ATyi · 2025-08-04
> >
> > Thank you for the response. However, I believe the current reply to **Issue 2** reveals some underlying problems.
> >
> > The author claims that the gradient itself is the motivation, but I find this argument problematic for several reasons:
> >
> > - In this end-to-end setup, the method is ultimately solving a mathematical optimization problem, and gradients do not always carry interpretable semantic meaning.
> > - The **objective function** is the true source of motivation, whereas the gradient is merely **a means to an end**.
> > - There is a gap between the story and the method. A method that better matches the narrative might involve: analyzing each step in the direction of the objective function, defining a new objective accordingly, and then computing its gradient—rather than analyzing directions directly in the gradient space as is currently done.
> >
> > Additionally, regarding the discussion of **Issue 4** and related work:
> >
> > - Some prior methods do redistribute the environment reward to others, not just intrinsic rewards. For example, the paper _"Learning to Incentivize Other Learning Agents"_ is one such case.
> > - The authors mention that their method allows agents to alternate between cleaning pollution and collecting apples, suggesting this leads to fairness. However, since pollution and apples are spatially separated, such task-switching incurs movement costs and reduces overall system efficiency. Thus, this strategy is suboptimal.
> > - The optimal approach is to divide labor and then distribute rewards afterwards. This is not only the most efficient but also relatively fair. It might be difficult to design a new method that outperforms this in "cleanup".

---

> > > ### Author Response · Authors · 2025-08-04
> > > **Response.**
> > >
> > > We appreciate the reviewer's response.
> > >
> > >
> > > ## Regarding "gradient and motivation" ##
> > >
> > > We do not claim that the gradient itself is the motivation. We agree with the reviewer that the objective function is the true source of motivation. We pursue that objective **via** gradient adjustment; optimizing the objective functions is our goal, and we achieve that goal through gradient adjustment.
> > >
> > > ---
> > >
> > > ## Regarding "gradients do not always carry interpretable semantic meaning"
> > >
> > > We agree that gradients do not always carry interpretable semantic meaning. Theoretically, we prove that FCGrad guarantees monotonically non-decreasing improvements, and that the expected individual returns (i.e., cooperation) across agents converge to the same value (i.e., fairness).
> > >
> > > ---
> > >
> > > ## Regarding " gap between the story and the method"
> > >
> > > The adjusted gradient can be interpreted as the gradient of a new objective, and this guides agents toward fairness and cooperation. We believe this matches the narrative presented in the paper.
> > >
> > > ---
> > >
> > > ## Regarding "redistribute the environment reward" ##
> > >
> > > **Reward is computed from what agents do in a given state. Even if rewards are redistributed, the fact that one agent sacrifices itself, for example, by cleaning pollution alone does not change.** The gifting methods we cited are also reward-redistribution approaches; therefore, we treat them as providing intrinsic rewards. This is why we focus exclusively on **extrinsic** rewards that reflect actual behavior. By the same logic, one might suggest maximizing the collective reward and redistributing it afterward. However, this approach entirely ignores individual objectives. Thus, we respectfully disagree that “the optimal approach is to divide labor and then distribute rewards.”
> > >
> > > That is why we introduce three \alpha-fairness metrics. Even in the \alpha=0 case, which ignores fairness, FCGrad performs as well as or better than the baselines; for \alpha=1 and \alpha = \infty, FCGrad outperforms them.
> > >
> > > ---
> > >
> > > ## Regarding "since pollution and apples are spatially separated, such task-switching incurs movement costs and reduces overall system efficiency. Thus, this strategy is suboptimal." ##
> > >
> > > Optimality depends on the metric. For collective reward (sum of all agents' rewards) alone, we agree that task-switching is not optimal (though FCGrad still performs similarly or better than the baselines). However, under \alpha-fairness, which balances cooperation and fairness, task-switching can be optimal. Generally speaking, these solutions all lie on the Pareto frontier, each corresponding to different weightings of the objectives.

---

> > > > ### Comment · Reviewer_ATyi · 2025-08-05
> > > >
> > > > Thank you for the further clarification. I am convinced by the methodological aspects of the paper; however, there are still some points with which I disagree.
> > > >
> > > > ---
> > > >
> > > > > The gifting methods we cited are also reward-redistribution approaches; therefore, we treat them as providing intrinsic rewards. This is why we focus exclusively on **extrinsic** rewards that reflect actual behavior.
> > > >
> > > > I believe there is a misunderstanding in the authors’ interpretation of the paper we are discussing (LIO). That paper assumes all agents are **self-interested and rational**. In terms of the framework design, what they propose is merely an interaction mechanism that allows agents to **exchange environment rewards**. These rewards are tangible and can be understood as currency, rather than intrinsic rewards.
> > > >
> > > > The outcomes observed in the _Escape Room_ and _Cleanup_ environments stem from this mechanism: one agent pays a certain amount of reward in exchange for the services of another. The convention formed in this multi-agent system essentially constitutes a **contract**.
> > > >
> > > > This reflects an important insight from **division of labor and cooperation in human society**: everyone performs their role, and the goods or services produced are exchanged via a universally accepted valuable item (e.g., money, or the environment reward) to facilitate cooperation.
> > > >
> > > > Therefore, the authors' interpretation ("we treat them as providing intrinsic rewards") is **incorrect**.
> > > >
> > > > > By the same logic, one might suggest maximizing the collective reward and redistributing it afterward. However, this approach entirely ignores individual objectives.
> > > >
> > > > Once again, I would like to emphasize that the approach ("maximizing the collective reward and redistributing it afterward") mentioned by the authors here is **forced**
> > > >
> > > > In contrast, the method in the LIO paper is grounded in game-theoretic principles. Each agent is self-interested, and the redistribution of rewards happens **voluntarily**, because the strategies under the mechanism are **incentive-compatible**. For example, in the _Escape Room_ scenario, without this reward redistribution mechanism, no agent would be motivated to open the door in the first place.
> > > >
> > > > Therefore, it is **not** "the same logic".
> > > >
> > > > ---
> > > >
> > > > I agree with most of the other parts. However, I still believe that the suboptimality in _Cleanup_, which the authors themselves acknowledge, is a **limitation** and may imply deeper underlying issues. For example, if optimality cannot be achieved in this setting, **does it suggest that there are certain types of tasks this algorithm is not well-suited for?** Are there tasks similar to _Cleanup_ where the proposed method would also perform poorly—such as _Escape Room_? How can we determine this boundary?

---

> > > > > ### Author Response · Authors · 2025-08-05
> > > > > **Response**
> > > > >
> > > > > We really appreciate the reviewer's active discussion.
> > > > >
> > > > >
> > > > > ## Regarding "The gifting methods we cited are also reward-redistribution approaches; therefore, we treat them as providing intrinsic rewards. This is why we focus exclusively on extrinsic rewards that reflect actual behavior."
> > > > >
> > > > > We believe the misunderstanding comes from the perspective on the assumptions—specifically, whether or not we treat the exchange of environment rewards as something explicit like money. We agree with the reviewer that under the assumption where money is allowed (i.e., additional incentives not strictly influenced by behaviors—such as money being independent of the goal of collecting apples), LIO can also be seen as doing some form of fairness. We also agree that this assumption is valuable for reflecting human society. One of our references, [1], which discusses formal contracts, assumes it.
> > > > >
> > > > > However, we consider a more primitive assumption of fairness: we only consider rewards that result from agents’ behavior toward the goal (e.g., rewards for collecting apples). We do not claim that this assumption is better than the one involving monetary exchange. Our work is based on a different perspective of fairness. We will make this clearer in the final paper.
> > > > >
> > > > > ----
> > > > >
> > > > > ## Regarding "maximizing the collective reward and redistributing it afterward": this is implemented as the Cen baseline in the LIO paper [2], where it is considered an approximate upper bound. They report Cen’s performance for Cleanup, but not for Escape Room. As seen in Figure 6 in [2], LIO performs worse than Cen.
> > > > >
> > > > > Regarding optimality, we believe the misunderstanding stems from different interpretations of the environment. We think the reviewer views Cleanup as a cooperative setting—in which case, division of labor would indeed be necessary for optimality. However, we view Cleanup as a mixed-motive setting, where simply maximizing apple collection is not necessarily optimal. There exist multiple optima that balance cooperation and fairness, which is captured by our evaluation metric, α-fairness.
> > > > >
> > > > > In addition, we would like to note that FCGrad also performs similarly to the optimal in terms of cooperation-only reward—what we refer to as “Col” and what [2] refers to as “Cen.” Therefore, we respectfully suggest that it is not accurate to say that our proposed method performs poorly in Cleanup.
> > > > >
> > > > > ---
> > > > >
> > > > > ## Regarding "Escape Room":
> > > > >
> > > > > In the Escape Room environment with parameters N = 2 and M = 1:
> > > > >
> > > > > - N = 2: There are two agents.
> > > > >
> > > > > - M = 1: The door opens only if at least one agent pulls the lever.
> > > > >
> > > > > Rewards:
> > > > >
> > > > > - An agent receives +10 for successfully reaching the door after the lever has been pulled.
> > > > >
> > > > > - −1 penalty is applied for each movement (e.g., from Start to Lever, or Lever to Door).
> > > > >
> > > > > The episode ends immediately when any agent reaches the door.
> > > > >
> > > > > Now, assuming the environment is treated as a cooperative setting—where we only care about maximizing total reward—the optimal solution is:
> > > > >
> > > > > (a) One agent pulls the lever, the other goes to the door:
> > > > >
> > > > > Lever agent: −1
> > > > >
> > > > > Door agent: +10
> > > > > → Collective return: +9, which is optimal in terms of total reward.
> > > > >
> > > > > However, when we incorporate fairness into the analysis—especially from the perspective of fair cooperation—the story changes.
> > > > >
> > > > > (b) Both agents go to the lever first, then both move to the door:
> > > > > Each agent pays −2 (two moves), and only one receives +10 before the episode ends.
> > > > > Assuming one agent reaches the door and receives +10, the final rewards are:
> > > > >
> > > > > Lever-only agent: −2
> > > > >
> > > > > Door agent: +8
> > > > > → Collective return: +6, which is lower than (a).
> > > > >
> > > > > Yet, if we evaluate both outcomes using the α-fairness metric with α = ∞ (which focuses on the minimum individual reward), then:
> > > > >
> > > > > (a): Minimum reward = −1
> > > > >
> > > > > (b): Minimum reward = +6
> > > > >
> > > > > From this perspective, (b) is more fair, even though it sacrifices some cooperative efficiency.
> > > > > Thus, if we care about both fairness and cooperation, (b) may be the preferred solution.
> > > > >
> > > > > Of course, if the environment allows multiple rounds of lever-door sequences within a single episode, then agents could take turns playing the Cooperator and Winner roles. In that case, alternating roles over time would yield both maximum collective return and perfect fairness.
> > > > >
> > > > > Again, we really appreciate the reviewer's active discussion and questions. We hope this helps clarify the misunderstanding.
> > > > >
> > > > > [1] Andreas Haupt et al., "Get It in Writing: Formal Contracts Mitigate Social Dilemmas in Multi-Agent RL," AAMAS 2023
> > > > > [2] Jiachen Yang et al., "Learning to Incentivize Other Learning Agents," NeurIPS 2020

---

> > > > > > ### Comment · Reviewer_ATyi · 2025-08-05
> > > > > >
> > > > > > - Issue 1.
> > > > > > 	- I see that these are different perspectives, I just hope for a clear explanation. Solved.
> > > > > > - "We think the reviewer views Cleanup as a cooperative setting—in which case, division of labor would indeed be necessary for optimality. However, we view Cleanup as a mixed-motive setting, where simply maximizing apple collection is not necessarily optimal."
> > > > > > 	- I also think this is a mixed-motive task, it's a social dilemma.
> > > > > > 	- In your previous response, you claimed that “For collective reward (sum of all agents' rewards) alone, we agree that task-switching is not optimal.” I thought we had reached a consensus.
> > > > > > 	- I also understand there are two metrics with a trade-off here, social welfare and fairness. I want to express that by incorporating a reward distribution mechanism, agents can achieve the highest capacity for both of these metrics, **at the same time.**
> > > > > > - Regarding "Escape Room"
> > > > > > 	- The strategy the authors propose is still the same as in Cleanup, which remains suboptimal compared to the LIO method. The specific reasons have been explained before.
> > > > > > 	- To be clear: the reward distribution mechanism can maximize social welfare without undermining fairness.
> > > > > >
> > > > > > I actually think the quality of this paper is good, but I believe the authors have not yet realized the limitations of their method. They have not discussed the characteristics of the tasks for which their method is applicable. These are the main reasons why I have not increased the score.

---

> ### Author Response · Authors · 2025-08-05
> **Response**
>
> - We are glad that many misunderstandings have been addressed.
>
>
> - We agree with the reviewer that reward distribution mechanisms can also address both metrics.
>
>
> - We acknowledge our limitation: SSD problems are originally aimed at resolving the tension between short-term individual incentives and long-term collective benefits in a temporally extended setting—where the collective return is the ultimate objective. Our formulation is suboptimal in that regard. We will clarify this point in the final version of the paper.
>
> Please let us know if there are any additional points that require clarification, so we can further address the reviewer’s concerns.
>
> Best,
>
> Authors

---

> > ### Comment · Reviewer_ATyi · 2025-08-08
> >
> > Based on the author’s reply to the third point, I believe the author still has not grasped my point. The cleanup task is not aimed at the collective return as the ultimate goal, but rather each agent optimizes its own individual reward in an agent-wise manner. However, there exists a mechanism that can lead selfish agents to achieve a situation where the collective return is optimal — this is precisely the charm of a good mechanism. The division of labor emerges naturally. I think the author still does not understand my previous reply.

---

> ### Author Response · Authors · 2025-08-08
>
> We apologize if our earlier statements came across as overly strong.
>
> When we referred to the “ultimate goal,” our intention was to describe the evaluation goal in most SSD studies, which is then reported as collective return in the evaluation results(e.g., Figures 4, 5, and 6 in [1]). Prior work typically evaluates performance based on collective return, in settings where individual incentives and long-term collective benefits are not naturally aligned. This misalignment creates tension, and the objective is to resolve it so that the total number of apples collected is maximized. In such cases — as in Cleanup — the absence of direct rewards for cleaning creates the social dilemma, which in turn hampers collective return. As the reviewer correctly noted, this goal is pursued as “each agent optimizes its own individual reward in an agent-wise manner.”
>
> As the reviewer also mentioned, existing approaches make use of reward redistribution (what we refer to as reward reconstruction). This assumes that redistributed rewards are real, tangible payoffs — akin to currency in real life — that can be used to address fairness issues, e.g., compensating an agent that only cleans. Under this assumption, division of labor naturally emerges. We find this approach compelling and elegant, and it has been the focus of much prior work.
>
> In contrast, as the reviewer acknowledged, our work adopts a different assumption. Rather than assuming the availability of a currency-like signal that is independent of an agent’s direct actions or the environment, we focus on a more primitive notion of fairness — one based solely on rewards obtained from actual state–action outcomes. This leads us to ask: if one agent cleans entirely while others only collect apples, is such an arrangement still acceptable in the absence of currency? In such a setting, one could argue that it would be fairer — and potentially better from another perspective — for all agents to share the cleaning task, even if that changes the collective reward outcome.
>
> We fully understand the reviewer’s concern that, under the traditional reward redistribution assumption, our method could be suboptimal in terms of collective reward. However, under our setting, there can be multiple optimal solutions depending on the trade-off between fairness and cooperation. For this reason, we evaluate using multiple α values in α-fairness to explore different points on the Pareto frontier.
>
> We will make sure to clearly explain this alternative assumption and the scope of our study in the final version.
>
> Thank you again for your thoughtful comments.
>
> ---
>
> [1] Jiachen Yang et al.,"Learning to Incentivize Other Learning Agents," NeurIPS 2020

---

> > ### Comment · Reviewer_ATyi · 2025-08-08
> >
> > Thank you for your follow-up reply. With each response, I feel we are reaching a greater consensus. However, I still have a few concerns.
> >
> > > We focus on a more primitive notion of fairness—one based solely on rewards obtained from actual state-action outcomes. This leads us to ask: if one agent cleans entirely while others only collect apples, is such an arrangement still acceptable in the absence of currency?
> >
> > In scenarios without reward redistribution, it is true that one agent cleaning entirely while others only collect apples is an unreasonable situation. Furthermore, **this is not an equilibrium.** Note that each agent optimizes their expected return, and RL involves exploration. Those who consistently clean will explore better strategies, such as occasionally eating an apple. I have no disagreement with this.
> >
> > However, note that this does not mean the author has corrected my view. What the author is saying and what I wanted to bring up are two different things.
> >
> > The author now seems to think that reward redistribution is a task setting, but in fact, **it is part of a method**, similar to how $\alpha$-fairness relates to the author's method. Therefore, this behavior can be compared with $\alpha$-fairness:
> >
> > - Let selfish agents spontaneously perform reward redistribution and observe their performance in cleanup.
> > - Let agents in the $\alpha$-fairness framework play and observe their performance in cleanup.
> >
> > The second method, the author's method, is less capable than the first. And authors agreed with this point.
> >
> > If the author wants to argue that reward redistribution is a setting, then we could also say that the $\alpha$-fairness framework is a setting, and one that does not conform to the assumption of self-interested and rational agents. **This does not seem "game theory." So why study this setting? It also does not help achieve a better optimum. The author's justification for this point and a discussion of the relevant literature would be very helpful.**
> >
> > My main concern before was not that the author misunderstood the LIO paper, but that the assumptions of the $\alpha$-fairness framework itself would lead to certain limitations. **Especially since the experiments use the cleanup benchmark environment, it is natural to need to compare it with other methods.**

---

> ### Author Response · Authors · 2025-08-08
>
> We sincerely thank the reviewer for actively engaging in the discussion and helping us refine our explanations.
>
> First, our work does not aim to find an equilibrium; rather, it seeks Pareto-optimal outcomes. Both equilibrium concepts (e.g., Nash, correlated equilibrium) and Pareto optimality are well-studied in game theory, but they are distinct. While an equilibrium focuses on stability under strategic deviations, Pareto optimality focuses on efficiency—solutions where no agent’s outcome can be improved without reducing another’s.
>
> Second, α-fairness in our paper is not a training framework but an evaluation metric. Our method does not directly use α-fairness during learning; we only use it to evaluate outcomes. As noted in the paper, we also report the α = 0 case (collective return), which corresponds to pure efficiency without fairness considerations. From the behavioral perspective, we understand that the reviewer may be referring to sharing the cleaning task in Cleanup. In that sense, policies learned with FCGrad may achieve slightly lower collective return than “selfish agents spontaneously performing reward redistribution.” In our view, this latter behavior is effectively equivalent to all agents directly optimizing collective reward, which is exactly what we report as the “Col” baseline in our experiments. Indeed, in Cleanup and Harvest, “Col” sometimes achieves a higher maximum collective return than FCGrad, and this is even more evident in Unfair Coin (Figure 3). However, for other α values, FCGrad outperforms “Col.” In addition, we think the reviewer's suggestion that directly incorporating α-fairness into the learning process could be an interesting direction. To the best of our knowledge, such approaches have been explored in bandit settings, but applying them in complex RL environments remains challenging. We will leave this as an avenue for future work.
>
> Finally, as mentioned above, game theory is not solely about equilibrium concepts. Our paper is about Pareto-optimal solutions, and α-fairness is used as a metric to evaluate them. While FCGrad may not achieve the highest performance for α = 0 (collective return), it outperforms baselines for α = 1 and α = ∞, which incorporate both task completion and fairness. We will make sure to clarify this distinction, as well as the role of α-fairness, more explicitly in the final version.
>
> Once again, we truly appreciate the reviewer’s active participation in this discussion, which has helped us better articulate the scope and positioning of our work.

---

> > ### Comment · Reviewer_ATyi · 2025-08-08
> >
> > > α-fairness in our paper is not a training framework but an evaluation metric. Our method does not directly use α-fairness during learning; we only use it to evaluate outcomes.
> >
> > The current discussion has spanned a long time, which led me to remember this part incorrectly. My previous comment was mistaken.
> >
> > As for the other points — the discussion about Pareto-optimality and equilibrium — I never had any objections before, and I was not arguing against the authors earlier. The author does not need to clarify those.
> >
> > The author has addressed my concerns. I have decided to raise my score and hope the author will add the discussion they promised to the paper.

---

> > > ### Author Response · Authors · 2025-08-08
> > >
> > > We sincerely thank the reviewer for the constructive and insightful discussion. We will include the promised clarifications in the final version. We greatly appreciate this kind of open, in-depth exchange, which we believe is essential for advancing the machine learning community.

---

### Official Review · Reviewer_V4P2 · 2025-07-01

**Clarity:** 3
**Significance:** 3
**Originality:** 2
**Rating:** 4
**Confidence:** 4

**Summary:**

The paper introduces a conflict-aware gradient adjustment-based multi-agent RL method to deal with both cooperation and fairness in mixed-motive games. The algorithm determines whether a conflict occurs through the inner product of the gradients of individual and collective objectives. If there is no conflict, FCGrad uses weighted sum gradient adjustment. If there is conflict, FCGrad places more weight on the individual (collective) gradient when the collective (individual) objective is greater, in order to ensure fairness. Experimental results show that the proposed algorithm outperforms the baselines in fairness in conventional mixed-motive game tasks.

**Questions:**

1. What are the specific differences between the algorithms FCGrad and PCGrad?

2. Do the reward functions for both individual and collective objectives come from feedback from the environment?

3. In addition to fairness metrics, how well does the algorithm perform on the task itself?

4. How scalable is the algorithm to multi-agent systems?

**Ethical Concerns:**

["NO or VERY MINOR ethics concerns only"]

**Final Justification:**

The authors addressed my concerns in rebuttal and added corresponding experiments and discussion, so I raised my score.

**Limitations:**

yes

**Quality:**

2

**Strengths And Weaknesses:**

Strengths：
1. The paper explicitly guarantees fairness through conflict-aware gradient adjustment. The theory is complete with synchronous and monotonic improvement guarantees for individual and collective objectives, and ultimate goal consistency.

2. The computational cost is only the gradient projection operation, so the computational overhead is low.

3. In typical mixed-motive games, the proposed algorithm outperforms the baselines in fairness metrics.

Weaknesses：
1. The paper only considers individual-collective conflicts, but potential goal conflicts between agents (individual-individual) are not involved.

2. The experimental indicators mainly reflect fairness, but lack measurement of "task completion" itself.

3. It has not been verified in large-scale complex heterogeneous multi-agent systems.

4. The proposed algorithm is highly dependent on the periodic occurrence of gradient conflicts, but periodic occurrence may not be guaranteed to always hold true in real-world.

5. The accuracy of gradient estimation for the two objectives may also be a potential problem.

---

> ### Author Rebuttal · Authors · 2025-07-31
>
> We appreciate the reviewer’s valuable comments and questions.
>
> Below, we provide our responses to the reviewer’s concerns and questions.
>
> ## [3-1] Regarding "individual-collective conflicts"
>
> We believe that individual–collective conflict implicitly includes individual–individual conflicts in the form of "one agent versus the others." For example, consider a three-agent case where the collective reward is defined as the average of individual rewards. If agent 1’s reward is lower than the average, i.e., (R1+R2+R3)/3 > R1, then it must be the case that the average of R2 and R3 is greater than R1, implying at least one agent is outperforming agent 1.
>
> ---
>
> ## [3-2] Regarding "lack measurement of 'task completion' itself"
>
> We respectfully clarify that the main results in Section 4 are based on $\alpha$-fairness, which simultaneously captures both "task completion" and "fairness." As discussed in the paper:
> - $\alpha=0$-fairness (equivalent to average return) captures only task completion;
> - $\alpha=1$-fairness (geometric mean) and $\alpha=\infty$-fairness (minimum individual return) incorporate both task performance and fairness, with increasing emphasis on fairness.
>
> ---
>
> ## [3-3] Regarding "large-scale complex heterogeneous multi-agent systems"
>
> Our current experiments focus on fairness and cooperation in environments that exhibit asymmetries, which we consider a form of heterogeneity. In addition, Cleanup and Harvest have been considered as fairly large-scale and complex environments. Scaling FCGrad to extremely large or structurally heterogeneous agent populations is an important challenge, and we agree it would be valuable future work.
>
> ---
>
> ## [3-4] Regarding "the periodic occurrence of gradient conflicts"
>
> We agree with the reviewer that the theoretical framework assumes the periodic occurrence of gradient conflicts. While this assumption may not always hold in real-world scenarios, our empirical results demonstrate that FCGrad performs well in the tested environments, where such conflicts naturally arise during learning. Investigating robustness under less frequent or non-periodic conflict patterns would be a promising direction for future research.
>
> ---
>
> ## [3-5] Regarding "The accuracy of gradient estimation"
>
> We agree with the reviewer that the quality of gradient estimation can impact learning. However, similar to general stochastic gradient methods, we mitigate this issue by using a sufficiently large number of samples when estimating gradients.
>
> ---
>
> ## [3-6] Regarding "the specific differences between FCGrad and PCGrad"
>
> First, PCGrad was originally proposed for multi-objective learning, whereas FCGrad is designed for achieving fair cooperation in multi-agent reinforcement learning. Second, PCGrad does not adaptively consider which gradient direction should be prioritized. It resolves conflicts by projecting each conflicting gradient onto the normal plane of the other and summing them (see Section 2.3 of our paper).
>
> In contrast, FCGrad adaptively selects which gradient to project based on the expected returns: the gradient associated with the lower expected return is projected onto the normal plane of the other. This mechanism promotes fairness by favoring the underperforming objective in conflict situations.
>
> ---
>
> ## [3-7] Regarding "reward functions"
>
> In our setup, each agent has access to all individual rewards provided by the environment. The collective reward is computed as the average of these individual rewards. Thus, both individual and collective rewards originate from the environment, with the latter being a derived aggregate.
>
> ---
>
> ## [3-8] Regarding "perform on the task itself"
>
> As described in our main results (Section 4), we use $\alpha$-fairness as a unified metric that captures both task performance (explicit reward) and fairness. Specifically, $\alpha = 0$ corresponds to average return (pure task performance), while $\alpha = 1$ and $\alpha = \infty$ incorporate increasing emphasis on fairness. FCGrad achieves competitive performance under $\alpha = 0$ and outperforms other baselines under $\alpha = 1$ and $\alpha = \infty$, indicating it balances both objectives effectively.
>
> ---
>
> ## [3-9] Regarding "How scalable is the algorithm to multi-agent systems"
>
> Since FCGrad is a decentralized algorithm, it is easy to scale, as reviewer ZK8G mentioned as one of our strengths (“2. It is a decentralized training approach which gives it scalability.”) We believe scaling to very large populations is a potential future direction.

---

> > ### Comment · Reviewer_V4P2 · 2025-08-05
> >
> > 1. The tasks (e.g., Cleanup and Harvest) have some metrics themselves, like the Utilitarian metric (U), Equality metric (E), Sustainability metric (S), and total contribution to the public good (P) [1]. How does your approach perform specifically on these metrics?
> >
> > 2. The Utilitarian metric (U), also known as collective return [1]. But for its formula, it may be different from your collective reward. Can you provide us the specific mathematical expressions for yours?
> >
> > 3. In your case ((R1+R2+R3)/3 > R1), it doesn't always occur in mixed-motive games; it can also occur in fully cooperative games, often due to poor credit allocation and the emergence of lazy agents. Improving the lazy agent's policy increases individual rewards and collective rewards simultaneously. This doesn't fully fit the mixed-motive game setting. Could you give us a more detailed explanation about the problem?
> >
> > [1] Hughes E, Leibo J Z, Phillips M, et al. Inequity aversion improves cooperation in intertemporal social dilemmas. NeurIPS 2018.

---

> > > ### Author Response · Authors · 2025-08-05
> > > **Response**
> > >
> > > We appreciate the reviewer’s valuable comments
> > >
> > > ## Regarding "some metrics"
> > >
> > > We have already provided the results in terms of the Utilitarian (U) and Equality (E) metrics.
> > >
> > > - Utilitarian metric (U):
> > > The metric used in their work is the sum of the collective rewards, whereas we report the average collective reward. These are essentially equivalent, differing only by a scaling factor (i.e., divided by the number of agents).
> > >
> > > - Equality metric (E):
> > > They use the Gini coefficient to measure equality. We have already reported this result in Section 4.5 (Table 1). Additionally, our α-fairness metric also incorporates a notion of equality, providing a broader evaluation.
> > >
> > > - Sustainability (S) and Public Good Contribution (P):
> > > Unfortunately, we did not log these metrics during training. Due to time constraints, we are unable to provide them at this time. However, we believe that our evaluation using α-fairness and additional fairness metrics sufficiently captures both cooperation and fairness aspects.
> > >
> > > ---
> > >
> > > ## Regarding "case ((R1+R2+R3)/3 > R1)"
> > >
> > > We agree that lazy agents can lead to such a case. In fact, this situation naturally arises in mixed-motive games.
> > > For example, in the Cleanup environment, when optimizing solely for collective reward, one agent may end up cleaning all the waste while the others collect apples (see Figure 4(b)). In this case, the agent performing the cleaning (R1) may receive no reward, while the average return across agents becomes higher than R1’s return.

---

> > > > ### Comment · Reviewer_V4P2 · 2025-08-09
> > > >
> > > > Thanks to the author for the response, I will raise my score. I hope the author will add these discussion to the paper.

---

### Official Review · Reviewer_mRtA · 2025-07-02

**Clarity:** 4
**Significance:** 3
**Originality:** 3
**Rating:** 5
**Confidence:** 4

**Summary:**

The paper presents a new gradient-based algorithm for MARL in which the returns of each agent are shaped to consider not only individual objectives, but also social welfare and fairness. The new algorithm balances each agent's learning to explicitly account for the individual and shared objectives so they are solved to maximize fairness. The paper includes two theorems that together form a theoretical basis for the learned policies to be optimal. Empirical evaluation includes three social dilemma domains and a variety of agents that are self-interested. The results show that the new method, FCGrad, enables the agents to learn a balanced policy that maintains fairness using various metrics.

**Questions:**

1. Did you perform any significance tests for the results in Table 1?

**Ethical Concerns:**

["NO or VERY MINOR ethics concerns only"]

**Final Justification:**

This is a good paper, and I am happy that the authors responded to the concerns of the reviewers to the extent that the paper has reached a unanimous positive rating.

**Limitations:**

It can be beneficial to have a brief discussion about the dual purpose of this research: can someone use the same paradigm as FCGrad to intentionally *decrease* fairness?

**Paper Formatting Concerns:**

No formatting issues.

**Quality:**

4

**Strengths And Weaknesses:**

Strengths:
+ This paper is one of the most precise and clear papers I have read recently. Well done!
+ The related work is comprehensive, follows a clear narrative, and provides context to understand the novelty of the new contribution.
+ Empirical evaluation is strong, with environments especially modified with asymmetries to emphasize the role of fairness in these domains.
+ The results support the overall superior performance of the new approach across several metrics.

Weaknesses:
- The main weakness is the theoretical analysis. Theorems are considered a part of the main paper, but the proofs are solely in the appendix, so they cannot be considered a fully sound contribution. Theorem 3.2 is a significant leap and requires additional preliminary explanations.
- In line 168, "FCGrad uses the weighted sum of two gradients: g = ..." I assume g and g_FCGrad are supposed to be the same, because if they're not, I'm not sure where the latter is used.
- In Table 1, results are bolded not just for the top-performing algorithm, but also for the second-best. This is confusing, especially when no significance is reported, so there's no way to verify these results are indeed "close".

---

> ### Author Rebuttal · Authors · 2025-07-31
>
> We appreciate the reviewer’s valuable comments and questions.
>
> Below, we provide our responses to the reviewer’s concerns and questions.
>
> ## [2-1] Regarding "Details of proof"
>
> Due to space limitations, we include only the interpretations of the statements in the main paper. We will provide a proof sketch in the final version.
>
> ---
>
> ## [2-2] Regarding "FCGrad uses the weighted sum of two gradients: g = ..."
>
> That is correct. We will revise g to g_fcgrad in line 168.
>
> ---
>
> ## [2-3] Regarding "Table 1: bold and significant tests"
>
> We will include the standard deviation in the final paper. The following table shows the mean and standard deviation of Jain and Gini indices for each algorithm. (We have added two more seeds for Harvest and Cleanup.)
>
> | Task (Metric)     | col       | ind       | iar       | wei       | pcg       | aga       | fcgrad (ours) |
> |-------------------|-----------|-----------|-----------|-----------|-----------|-----------|----------------|
> | Coin (Gini)       | 0.47±0.00 | 0.51±0.08 | 0.12±0.02 | 0.05±0.00 | 0.04±0.01 | 0.29±0.02 | 0.01±0.01     |
> | Coin (Jain)       | 0.53±0.00 | 0.50±0.08 | 0.94±0.02 | 0.99±0.00 | 0.99±0.00 | 0.75±0.02 | 1.00±0.00   |
> | Cleanup (Gini)    | 0.50±0.12 | 0.49±0.15 | 0.44±0.21 | 0.23±0.06 | 0.36±0.21 | 0.32±0.06 | 0.21±0.11     |
> | Cleanup (Jain)    | 0.50±0.12 | 0.53±0.18 | 0.61±0.24 | 0.83±0.06 | 0.69±0.24 | 0.74±0.08 | 0.84±0.12     |
> | Harvest (Gini)    | 0.24±0.13 | 0.12±0.02 | 0.09±0.01 | 0.16±0.06 | 0.09±0.01 | 0.10±0.03 | 0.08±0.05     |
> | Harvest (Jain)    | 0.82±0.15 | 0.95±0.02 | 0.97±0.00 | 0.91±0.07 | 0.97±0.01 | 0.96±0.02 | 0.97±0.03     |
>
> The following table shows the p-values from Student’s t-test comparing FCGrad to each baseline:
>
> | Task            | col   | ind   | iar   | wei   | pcg   | aga   |
> |-----------------|-------|-------|-------|-------|-------|-------|
> | Coin (Gini)     | 0.00  | 0.01  | 0.01  | 0.00  | 0.06  | 0.00  |
> | Coin (Jain)     | 0.00  | 0.01  | 0.04  | 0.01  | 0.11  | 0.00  |
> | Cleanup (Gini)  | 0.01  | 0.04  | 0.10  | 0.74  | 0.25  | 0.12  |
> | Cleanup (Jain)  | 0.00  | 0.05  | 0.12  | 0.90  | 0.29  | 0.15  |
> | Harvest (Gini)  | 0.04  | 0.13  | 0.55  | 0.03  | 0.46  | 0.42  |
> | Harvest (Jain)  | 0.07  | 0.22  | 0.99  | 0.11  | 0.91  | 0.68  |
>
> Note that these results reflect fairness only, without considering task performance. In the Unfair Coin Game, FCGrad achieves higher fairness metrics than all baselines, with strong statistical significance. For Cleanup, FCGrad outperforms all except wei, which achieves similar fairness. In Harvest, most algorithms except col show comparable fairness to FCGrad. Even though FCGrad does not always outperform others in raw fairness scores, it achieves statistically significant improvement in terms of α-fairness (please refer to our response to Reviewer ZK8G03 [1-2]).
>
>
>
> ## [2-4] Regarding "a brief discussion about the dual purpose of this research: can someone use the same paradigm as FCGrad to intentionally decrease fairness?"
>
> Thank you for the helpful and meaningful suggestion. We would like to clarify that FCGrad does not directly optimize for fairness as an explicit objective; rather, fairness emerges as a consequence of balancing individual and collective objectives. Therefore, we do not believe FCGrad can be trivially repurposed to intentionally decrease fairness.
> In fact, an extreme case of not considering fairness at all would be optimizing only for the collective objective, i.e., the Col baseline, which may lead to highly imbalanced outcomes despite maximizing the overall return.
>
> That said, as the reviewer suggests, there are certainly ways an agent could attempt to manipulate fairness, such as by maximizing its own reward while actively minimizing others’. Detecting and preventing such adversarial behaviors presents an important and underexplored direction for future work in fairness-aware multi-agent systems.

---

### Official Review · Reviewer_ZK8G · 2025-07-03

**Clarity:** 3
**Significance:** 2
**Originality:** 2
**Rating:** 4
**Confidence:** 4

**Summary:**

In summary, this paper introduces the idea of fair and conflict-aware gradient adjustments for the updates of policies which can achieve cooperation and fairness. It outperforms current baselines in terms of fairness in the social welfare in mixed-motive multi-agent sequential social dilemmas.

They propose a fair and conflict-aware gradient adjustment method (FCGrad) which guarantees a monotonic non-decrease of both individual and collective objectives, while preserving fairness across each individuals objective.
FCGrad, which works in a decentralized model, uses two value functions, for the individual and for the collective. This imposes the assumption that there is a shared global reward. They then find the gradients of the individual and the collective objectives respectively. They then check the sign inner product of these gradients. If positive, they use a weighted sum of the two gradients, and if it is negative, there is a conflict. It will then place more weight on the gradient that is lower to ensure fairness. They do this by projecting the individual or collective gradients onto the normal plane of another gradient vector, the exact form is described in equation (2).

This is shown both theoretically and empirically.

The empirical results are shown in the unfair coins, cleanup and harvest environments, which are slightly tweaked versions of previously used versions of these environments.
They compare with 6 baselines: Collective reward optimization (Col), Individual reward optimization (Ind), inequity aversion reward restructuring (IA), FCGrad without conflict handling, PCGrad, and Altruistic Gradient Adjustment. They also include a hyperparameter for the weighting of FCGrad, they show the choice of this value is significant to the method working, and is environment dependent.

Overall they show that FCGrad (Their method) results in being in the top two fairest outcomes across the methods over multiple fairness metrics, Gini and Jain. (Except Jain in Harvest, but it still close to the top of the performance)

**Questions:**

1.	While it is not the focus of this study. What do you think the outcome of agents trained with this type of reward being mixed into an environment with other training regimes?

2.	Would they be exploited?

3.	Would they be able to coordinate with different behaviours that are off policy from their own?

**Ethical Concerns:**

["NO or VERY MINOR ethics concerns only"]

**Final Justification:**

I believe that the authors have addressed my concerns.
Taking into account this and the other reviews, I will be keeping my score for possible acceptance.

**Limitations:**

Yes.

**Paper Formatting Concerns:**

Nothing major.

**Quality:**

3

**Strengths And Weaknesses:**

Strengths:

1.	This is an interesting take on how to achieve fairness in these mixed-motive environments.

2.	It is a decentralized training approach which gives it scalability.

3.	It compares to a number of known algorithms for fairness over multiple well known environments.

Weaknesses:

1.	It does describe how the model is set up, however, I believe a diagram would help a lot, even it were in the appendix.

2.	The reported results only being over 4 random seeds makes it seem like its still possible that these are specific ones which it works particularly well for. Increasing the number of runs could help.

---

> ### Author Rebuttal · Authors · 2025-07-31
>
> We appreciate the reviewer’s valuable comments and questions.
>
> Below, we provide our responses to the reviewer’s concerns and questions.
>
>
> ## [1-1] Regarding "a diagram for the model"
> We will include a diagram describing our model in the appendix of the final paper. Thank you for the helpful suggestion.
>
> ---
>
> ## [1-2] Regarding "Increasing the number of runs"
> We have added two more seeds (6 in total) for Cleanup and Harvest.
>
> The following table shows the performance of FCGrad and the baselines over 6 seeds:
>
> | Task                          | col      | ind      | iar      | wei       | pcg      | aga      | fcgrad (ours) |
> |------------------------------|----------|----------|----------|-----------|----------|----------|----------------|
> | Harvest (Geomean, α=1)       | 45.3±20.7 | 20.4±2.3 | 21.2±2.3 | 47.1±11.9 | 20.2±2.5 | 35.4±6.9 | **64.9±5.4**   |
> | Harvest (Min, α=∞)           | 23.3±20.2 | 12.6±1.4 | 14.3±0.2 | 29.3±13.2 | 13.8±0.7 | 24.8±1.4 | **49.5±12.4**  |
> | Cleanup (Geomean, α=1)       | 49.6±21.4 | 2.5±2.5  | 3.0±2.4  | 57.9±6.9  | 24.4±26.2| 3.0±2.5  | **76.0±15.6**  |
> | Cleanup (Min, α=∞)           | 12.0±10.3 | 0.01±0.00| 0.02±0.02| 31.1±11.5 | 11.4±22.8| 0.01±0.00| **43.6±27.6**  |
>
>
> The following table shows the p-values from Student’s t-test:
>
> | Task                        | col   | ind   | iar   | wei   | pcg   | aga   |
> |----------------------------|-------|-------|-------|-------|-------|-------|
> | Harvest (Geomean, α=1)     | 0.04  | 0.00  | 0.00  | 0.02  | 0.00  | 0.00  |
> | Harvest (Min, α=∞)         | 0.02  | 0.00  | 0.00  | 0.01  | 0.03  | 0.01  |
> | Cleanup (Geomean, α=1)     | 0.07  | 0.00  | 0.00  | 0.05  | 0.01  | 0.00  |
> | Cleanup (Min, α=∞)         | 0.05  | 0.02  | 0.02  | 0.38  | 0.09  | 0.02  |
>
>
> As shown in both tables, FCGrad consistently outperforms the baselines in terms of α-fairness (α=1 and α=∞), with statistical evidence. The only exception is the "wei" baseline in Cleanup (Min performance), which can exhibit high variance as α increases in α-fairness. However, FCGrad (43.6) still achieves better performance than Wei (31.1).
>
> ---
>
> ## [1-3] Regarding "Regarding the expected behavior of agents trained under different regimes"
>
> We appreciate the reviewer’s thoughtful questions regarding the robustness of FCGrad agents when interacting with agents trained using different regimes or exhibiting off-policy behaviors.
>
> We would first like to clarify that the theoretical guarantees of FCGrad (e.g., monotonic improvement and fairness) assume that all agents adopt the FCGrad update rule. Nevertheless, even in mixed training regimes, our design encourages coordination rather than exploitation due to the structure of the collective reward.
>
> In particular, since the collective objective is designed to yield higher returns than purely individual optimization (i.e., col > ind), agents trained with other regimes are naturally incentivized to engage in coordination. If an agent trained with a different policy attempts to exploit FCGrad agents by behaving selfishly, it will often find that its own individual reward remains limited, potentially perceiving this as an unfair outcome. Over time, it would thus be driven toward more cooperative behavior to access the benefits of the collective reward.
>
> Conversely, if an FCGrad agent perceives that others are not cooperative or are unable to adapt, it will naturally shift toward optimizing the individual objective. This dynamic ensures that coordination is pursued when beneficial, but exploitation is not blindly tolerated. In this way, FCGrad agents exhibit a degree of robustness when interacting with off-policy behaviors by adapting their gradient direction based on observed reward feedback.
>
> We believe this structure makes FCGrad well-suited for mixed-agent scenarios, and we thank the reviewer for raising this important point.

---

> > ### Comment · Reviewer_ZK8G · 2025-08-05
> >
> > Thank you for your answers.
> >
> > I believe that the authors have addressed my concerns.
> > Taking into account this and the other reviews, I will be keeping my score.

---

### Decision · Program_Chairs · 2025-09-17

**Decision:**

Accept (spotlight)

**Comment:**

The paper is technically solid, makes an original contribution to fairness-aware MARL, and generated thoughtful discussion that improved its clarity and positioning. The final version should incorporate the promised clarifications regarding theoretical proofs, fairness definitions, connections to redistribution methods, and the scope of applicability.